# On Optimal Steering to Achieve Exact Fairness

**Mohit Sharma**[*]
Department of Computer Science
IIIT Delhi, India
mohits@iiitd.ac.in

**Amit Jayant Deshpande**
Microsoft Research India
amitdesh@microsoft.com

**Chiranjib Bhattacharyya**
Department of Computer Science and Automation
Indian Institute of Science, Bengaluru, India
chiru@iisc.ac.in

**Rajiv Ratn Shah**
Department of Computer Science
IIIT Delhi, India
rajivratn@iiitd.ac.in

## Abstract

To fix the 'bias in, bias out' problem in fair machine learning, it is important to steer feature distributions of data or internal representations of Large Language Models (LLMs) to *ideal* ones that guarantee group-fair outcomes. Previous work on fair generative models and representation steering could greatly benefit from provable fairness guarantees on the model output. We define a distribution as *ideal* if the minimizer of any cost-sensitive risk on it is guaranteed to have exact group-fair outcomes (e.g., demographic parity, equal opportunity)—in other words, it has no fairness-utility trade-off. We formulate an optimization program for optimal steering by finding the nearest *ideal* distribution in KL-divergence, and provide efficient algorithms for it when the underlying distributions come from well-known parametric families (e.g., normal, log-normal).

Empirically, our optimal steering techniques on both synthetic and real-world datasets improve fairness without diminishing utility (and sometimes even improve utility). We demonstrate affine steering of LLM representations to reduce bias in multi-class classification, e.g., occupation prediction from a short biography in Bios dataset (De-Arteaga et al.). Furthermore, we steer internal representations of LLMs towards desired outputs so that it works equally well across different groups.

## 1 Introduction

The importance of clean or *ideal* data in fair machine learning cannot be overstated. The principle of *bias in, bias out* is widely recognised as a root cause of unfair outcomes in ML systems [16, 52, 62, 26]. Models trained on biased data tend to learn, perpetuate, and often amplify such biases. Importantly, the problem of biased data extends beyond training: fairness-constrained training on biased data does not guarantee fairness on (unbiased) test sets, and post-processing using biased validation data fails to ensure fairness at deployment. Moreover, fairness audits based on biased assessment data can be misleading and difficult to reverse [11, 4].

Fairness metrics such as demographic parity and equal opportunity are inherently functions of both the model and the data distribution. Thus, unfair outcomes may be addressed either by adjusting the model or by altering the data distribution. While prior work on fair in-processing aims to construct an *ideal* model under fairness constraints [1, 29], our work focuses instead on identifying an *ideal* data distribution that supports fairness. In this respect, our approach aligns most closely with the fair pre-processing literature.

---

[*]Part of the work was done when the author was an intern at Microsoft Research India.

39th Conference on Neural Information Processing Systems (NeurIPS 2025).

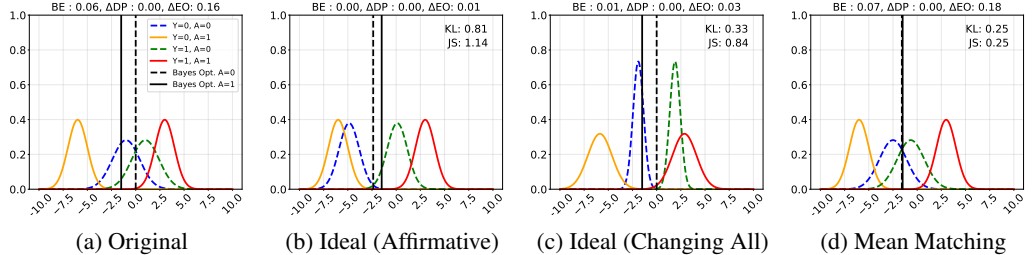

(a) Original   (b) Ideal (Affirmative)   (c) Ideal (Changing All)   (d) Mean Matching

Figure 1: Comparison of different interventions for changing Data Distributions for Exact Fairness. Figure (1a) captures the original distribution, its Bayes error (BE), and the unfairness differences ($\Delta$DP and $\Delta$EO). In Figure (1b), we only change the under-privileged group using Corollary 4.2, and in Figure (1c) we change all four subgroups using Proposition 4.3. Finally, in Figure (1d), we match the means of the two groups. Figures (1b) and (1c) show that it is possible to construct 'ideal' distributions that are close to the given distribution where both the BE and $\Delta$DP/$\Delta$EO are small.

Early work by Kamiran and Calders [43] introduced heuristic reweighting methods for binary classification, adjusting the weight of instances from class $i$ and group $a$ using $\Pr\left(\text{class } i\right) \cdot \Pr\left(\text{group } a\right)/\Pr\left(\text{class } i, \text{group } a\right)$. However, this approach ignores the feature distribution and offers no fairness guarantees when combined with accuracy maximisation. Calmon et al. [18] framed fair pre-processing as an optimisation problem that balances distance to the original distribution with group and individual fairness constraints. While convex in some settings, their approach may be infeasible when group and individual fairness are incompatible [34].

Other work explores alternative mechanisms: Jiang and Nachum [41] address label bias through reweighting; Plečko and Meinshausen [60], Plečko et al. [61] propose fair adaptation methods based on causal models; and Xiong et al. [75] reformulate pre-processing as a large-scale mixed-integer program, solved via a cutting-plane method. Fair pre-processing remains widely used in practice, often integrated into fairness toolkits alongside in- and post-processing methods [7, 10]. Despite its heuristic nature, the reweighing method of Kamiran and Calders [43] continues to perform well in mitigating bias across standard benchmarks [65, 12, 75].

A common goal of all fair pre-processing methods is to find an *ideal* data distribution close to the given distribution so that any downstream model trained on it must have guaranteed fairness. A stronger requirement that this should hold for downstream models optimized for multiple tasks leads to impossibility results [47]. If all downstream classifiers are required to be fair, then the group-wise distributions must be nearly identical, which is absurd. Thus, we restrict our downstream models only to Bayes optimal classifiers for cost-sensitive risks. Our first contribution is to formally define an *ideal* distribution as the one where the Bayes optimal classifier for any cost-sensitive risk satisfies exact fairness (e.g., satisfies equal opportunity perfectly).

Bayes optimal classifier maximizes accuracy on a given distribution, and have been an important object of study in statistical machine learning [28]. Fair Bayes optimal classifier maximizes accuracy subject to fairness constraints, and its mathematical characterization for binary fair classification has been important in fair classification [54, 24, 20, 78]. Blum and Stangl [13] introduce a data bias model that injects under-representation and label bias in an original unbiased distribution to create biased data. They show that, for a stylized distribution under some conditions, the fair Bayes optimal classifier on the biased distribution recovers the Bayes optimal classifier on the original unbiased distribution. Their unbiased distribution is *ideal* by construction, i.e., the Bayes optimal classifier on their unbiased distribution is guaranteed to be perfectly fair. Sharma and Deshpande [64] extend this observation to general hypothesis classes and distributions beyond the stylized setting of Blum and Stangl [13]. Blum et al. [14] study fair Bayes optimal classifier whether its accuracy is robust to malicious corruptions in data distribution. In contrast to these results, our focus is not on finding the *ideal* classifier but on finding the nearest *ideal* distribution.

By definition, our *ideal* distribution has no trade-off between accuracy (or cost-sensitive utility) and fairness. If we find an *ideal* distribution close to our original distribution, we can steer our distribution towards reducing fairness-accuracy trade-off. Moreover, if the *ideal* distribution offers better accuracy, it suggests that we can steer our distribution to improve both accuracy and fairness simultaneously. Fig. 1 gives such an example where the pre-processing of [43], in contrast, leaves the

original distribution unchanged and our method provides a direction to steer to distribution towards better accuracy and fairness simultaneously.

Dutta et al. [30] characterize a similar objective using Chernoff Information (see Cover [25]) and formulate an optimization program to find the nearest distribution in KL-divergence on which the Chernoff Information gap between two group-conditional feature distributions vanishes. Their optimization problem is not known to be efficiently solvable and the fairness guarantees in terms of Chernoff Information gap does not translate easily to standard fairness metrics such as demographic parity, equal opportunity etc. In contrast, we formulate an optimization problem to find the nearest *ideal* distribution in KL-divergence to given distribution and give efficient algorithms to solve it for various parametric families of distributions. To put our results to the test, we also apply our interventions to steer the representations of an LLM [72, 36] for fair multi-class classification [27, 67] and emotion steering [79]. We are able to improve the effectiveness of steering without significantly affecting the accuracy and are able to demonstrate how our notion of ideal distributions can help guide interventions for practical applications.

For completeness, we want to also make the reader aware of a long line of work on fair representation learning where the data transformations can map the distributions to another space [77, 50, 53, 49, 21]. Our work is not directly related but can potentially be used to refine fair representations to achieve provable and exact fairness guarantees.

## 1.1 Our Results

We summarize our key contributions as follows.

- We define *ideal* distribution (Definition 3.1) for fair classification as the one on which the Bayes optimal classifier for any cost-sensitive risk satisfies exact fairness (e.g., exact demographic parity, exact equal opportunity).

- When group and class-conditioned distributions belong to well-known parametric families of distributions (e.g., Gaussian, log-normal), we can succinctly rewrite the property of being an *ideal* distribution as a parametric condition (Proposition 3.3).

- The problem of finding the nearest *ideal* distribution to a given distribution is intractable in general. We formulate it as an optimization problem of KL-divergence subject to the parametric conditions required for being *ideal* (Section 4). As stated, it is a non-convex optimization problem (Proposition 4.3) but we show how to solve it efficiently. We also provide a closed form optimal solution in special cases (Theorem 4.1, Corollary 4.2). When better accuracy is achievable on the nearest *ideal* distribution, it suggests a direction to steer the original distribution in to improve both accuracy and fairness.

- To illustrate the effect of different interventions, we show the effect of using Affirmative Action (Corollary 4.2), changing all subgroups (Proposition 4.3) and matching the means of sensitive groups on different univariate distribution where we can compute the Bayes Optimal Group-aware classifiers analytically in Figures 1-2.

- To demonstrate how our guarantees can aid practical applications we also performs experiments to steer LLM representations to reduce bias in multi-class classification (Figure 3) and effective group-level emotion steering (Figure 4).

## 2 Problem Setup and Preliminaries

Let $(X, A, Y)$ be a random data point from a joint distribution $D$ over $\mathcal{X} \times \mathcal{A} \times \mathcal{Y}$, where $\mathcal{X}, \mathcal{A}, \mathcal{Y}$ denote the sets of features, sensitive attributes, and class labels, respectively. Let $q_{ia} = \Pr(Y = i, A = a)$ and $P_{ia}$ denote the distribution $X \mid Y = i, A = a$ with the probability density $p_{ia}(x) = \Pr(X = x \mid Y = i, A = a)$. When $P_{ia}$'s come from parametric families of distributions, we assume $\mathcal{X} = \mathbb{R}^d$. We work with the following well-known definitions of fairness in classficiation [31, 38, 6].

**Definition 2.1.** For exact fairness, in the case of multiple classes ($|\mathcal{Y}| > 2$) and multiple protected groups ($|\mathcal{A}| > 2$), a classifier $h : \mathcal{X} \times \mathcal{A} \to \mathcal{Y}$ satisfies:

1. *Demographic Parity* [31] if the positive rate for a class across groups is zero, i.e., $\max_{a,a' \in \mathcal{A}} |\Pr(h(X, A) = y | A = a) - \Pr(h(X, A) = y | A = a')| = 0, \forall y \in \mathcal{Y}$,

2. *Equal Opportunity* [38] if the true positive rates for a class across groups is zero, i.e., $\max_{a,a' \in \mathcal{A}} |\Pr(h(X, A) = y | Y = y, A = a) - \Pr(h(X, A) = y | Y = y, A = a')| = 0, \forall y \in \mathcal{Y}$.

These lead to quantitative metrics of unfairness, e.g., $\Delta_{\text{DP},y}(h, D)$ denotes the absolute value of difference between $\Pr(h(X, A) = y | A = a)$ and $\Pr(h(X, A) = 1 | A = a')$. Similarly, $\Delta_{\text{EO},y}(h, D)$ denotes the absolute value of difference between $\Pr(h(X, A) = y | Y = y, A = a)$ and $\Pr(h(X, A) = y | Y = y, A = a')$. Whenever we are dealing with binary classification, we will omit the use of $y$ in the subscript.

We consider group-aware classifiers. For binary classification tasks, we are particularly interested in threshold classifiers $h_t(x, a)$ that apply a group and feature dependent threshold $t(x, a)$ to the class probability of an example: $h_t(x, a) = \mathbb{I}(\eta(x, a) \geq t(x, a))$ where $\eta(x, a) = \Pr(Y = 1 | X = x, A = a)$. It is well-known that the Bayes optimal classifier for a given distribution has the form $t(x, a) = 1/2$ [28]. For a cost matrix $C \in \mathbb{R}^{2 \times 2}$ and the associated cost sensitive loss $l_C$, the Bayes optimal classifier is defined as $\mathbb{I}(\eta(x, a) \geq t_C)$, for a threshold $t_C = (c_{10} - c_{00})/(c_{10} - c_{00} + c_{01} - c_{11}) \in [0, 1]$, where $c_{ij}$ denote the entries of the cost matrix $C \in \mathbb{R}^{2 \times 2}$ [33, 63, 46, 66].

## 3 Ideal Distributions for Fair Classification

We define a data distribution as *ideal* when minimizing any cost-sensitive risk on it is guaranteed to give exact fairness (e.g., demographic parity, equal opportunity). In practice, downstream models trained on a distribution are typically optimized for some performance or utility metric that may not be known in advance. Our definition of ideal distribution allows the flexibility to choose any cost-sensitive risk as the performance metric for downstream models and still gives exact fairness guarantee for any optimal model downstream.

**Definition 3.1.** Let $\mathcal{H}$ be a hypothesis class of group-aware classifiers $h : \mathcal{X} \times \mathcal{A} \to \mathcal{Y}$ and let $\Delta(h, D)$ be a given unfairness metric, e.g., $\Delta_{\text{DP}}, \Delta_{\text{EO}}$. Given a distribution $D$ over $\mathcal{X} \times \mathcal{A} \times \mathcal{Y}$ and a cost-sensitive risk $C \in \mathbb{R}^{|\mathcal{Y}| \times |\mathcal{Y}|}$, let $h_C^* = \underset{h \in \mathcal{H}}{\text{argmin}} \Pr(\ell_C(h(X, A), Y))$. We call $D$ an *ideal distribution* if $\Delta(h_C^*, D) = 0$, for all $C \in \mathbb{R}^{|\mathcal{Y}| \times |\mathcal{Y}|}$.

Examples of fairness metrics include Demographic Parity, Equal Opportunity, and Equalized Odds (Definition 2.1 and Hardt et al. [38]), and examples of cost-sensitive risk include the usual $0 - 1$ loss and different performance metrics which are functions of the confusion matrix metrics [33, 46, 66].

Our definition gets around the impossibility theorems about fair representation for multiple tasks [47]. However, we need to be careful of two things. First, our definition should not be too restrictive to just force the group-conditioned distributions to be similar or identical, as that would be impractical. Second, we need an efficient and equivalent way of expressing the constraint of being ideal. We show how to express it as a parametric condition when the group and class-conditioned distributions belong to certain well-known parametric families of distributions. This helps in checking if a given distribution is ideal, and otherwise, finding its nearest ideal distribution.

### 3.1 Parametric Conditions for Ideal Distributions

Borrowing a simple setup of parametric distributions from previous work on fair machine learning [59], we assume that the class and group-conditioned feature distributions $X \mid Y = i, A = a$ belong to a parametric family of distributions, e.g., univariate or multivariate Gaussians, log-normal. In that case, we show that the property of being *ideal* (Definition 3.1) can be equivalently expressed as certain parametric conditions. To begin, we will first show the set of parametric conditions for multivariate normal distributions and a multi-class, multi-attribute setting.

**Proposition 3.2.** *Let $(X, Y, A)$ denote the features, class label, and group membership, respectively, of a random data point from any data distribution $D$ with $q_{ia} = \Pr(Y = i, A = a)$, for $i \in \mathcal{Y}$*

and $a \in \mathcal{A}$. Let $X|Y = i, A = a \sim \mathcal{N}(\mu_{ia}, \Sigma_{ia})$ be multivariate Normal distributions with mean $\mu_{ia} \in \mathbb{R}^d$ and covariance matrix $\Sigma_{ia} \in \mathbb{R}^{d \times d}$, for $i \in \mathcal{Y}$ and $a \in \mathcal{A}$. If the means $\mu_{ia}$ and the covariance matrices $\Sigma_{ia}$ satisfy

$$\Sigma_{ia}^{-1/2}(\mu_{ia} - \mu_{ja}) = \Sigma_{ia'}^{-1/2}(\mu_{ia'} - \mu_{ja'}) \quad and$$
$$\Sigma_{ia}^{1/2}\Sigma_{ja}^{-1}\Sigma_{ia}^{1/2} = \Sigma_{ia'}^{1/2}\Sigma_{ja'}^{-1}\Sigma_{ia'}^{1/2} \quad and \quad \frac{q_{ia}}{q_{ja}} = \frac{q_{ia'}}{q_{ja'}}, \; \forall i,j \in \mathcal{Y}, a, a' \in \mathcal{A},$$

*then the group-aware Bayes optimal classifier on $D$ satisfies equal opportunity.*

The proof for Proposition 3.2 is given in Section A of the Appendix. When we are dealing with binary class labels and group membership, we can show a necessary and sufficient condition for univariate normal distributions.

**Proposition 3.3.** *Let $(X, Y, A)$ denote the features, binary class label, and binary group membership, respectively, of a random data point from any data distribution $D$ with $q_{ia} = \Pr(Y = i, A = a)$, for $i \in \{0, 1\}$ and $a \in \{0, 1\}$, and let $X|Y = i, A = a \sim \mathcal{N}(\mu_{ia}, \sigma_{ia}^2)$ be univariate normal distributions, for $i \in \{0, 1\}$ and $a \in \{0, 1\}$. Then the distribution $D$ is* ideal *for equal opportunity (see Definition 3.1) if and only if*

$$\frac{\mu_{01} - \mu_{11}}{\sigma_{11}} = \frac{\mu_{00} - \mu_{10}}{\sigma_{10}}, \quad \frac{\sigma_{11}}{\sigma_{01}} = \frac{\sigma_{10}}{\sigma_{00}}, \quad \frac{q_{10}}{q_{00}} = \frac{q_{11}}{q_{01}}.$$

Proposition 3.3 shows that our parametric condition is equivalent to $\Delta_C(h_C^*, D) = 0$, for all cost matrices $C \in R^{2 \times 2}$. When we use a fixed cost matrix for 0-1 loss, and consider the Bayes optimal classifier in Proposition 3.2, our parametric condition is sufficient but not always necessary. However, the same condition ensures the Bayes optimal classifier to satisfy multiple fairness criteria simultaneously, viz., demographic parity, equal opportunity, equalized odds.

The proof for Proposition 3.3 is given in Section 1 of the supplementary material. It is interesting to note that the same parametric conditions imply that the Bayes optimal classifier on the corresponding distribution simultaneously satisfies multiple fairness criteria, viz., demographic parity, equal opportunity, and equalized odds. Moreover, the same condition works for both univariate Gaussian and log-normal distributions. Using our proof technique, it is easy to derive similar conditions for other parametric families too. We now present a small proposition to link our intervention to a widely used reweighing intervention in the fairness literature [43].

*Remark* 3.4. (Our conditions imply Kamiran and Calders [43] Intervention) Kamiran and Calders [43] reweighing method essentially reweighs $q_{ia}$ by a multiplicative factor of $\Pr(Y = i)\Pr(A = a)/\Pr(Y = i, A = a)$. Let us call the resulting probabilities $\tilde{q}_{ia}$. Using $\Pr(Y = i, A = a) = q_{ia}$, $\Pr(Y = i) = \sum_{a \in \mathcal{A}} q_{ia}$ and $\Pr(A = a) = \sum_{i \in \mathcal{Y}} q_{ia}$, we get $\tilde{q}_{ia} \propto q_{ia}(\sum_{a \in \mathcal{A}} q_{ia})(\sum_{i \in \mathcal{Y}} q_{ia})/q_{ia}$. Hence, $\tilde{q}_{ia}/\tilde{q}_{ja} = \tilde{q}_{ia'}/\tilde{q}_{ja'} = \sum_{a \in \mathcal{A}} q_{ia}/\sum_{a \in \mathcal{A}} q_{ja}, \forall i, j \in \mathcal{Y}$ and $a, a' \in \mathcal{A}$. It is the same condition on $q_{ia}$'s stated in Proposition 3.2. Thus, our result can be thought of as a second stage pre-processing of $P_{ia}$ distributions after applying the reweighing of Kamiran and Calders [43] to $q_{ia}$'s in the first stage.

*Remark* 3.5. (Limitation of [30]) As an interesting consequence, our conditions on $\mu_{ia}$ and $\Sigma_{ia}$ imply $D_{\mathrm{KL}}\left(\tilde{P}_{00}||\tilde{P}_{01}\right) = D_{\mathrm{KL}}\left(\tilde{P}_{10}||\tilde{P}_{11}\right)$. When the classes are balanced, the error rate of the Bayes optimal classifier on group $A = a$ in $\tilde{D}$ equals $0.5(1 - d_{\mathrm{TV}}(\tilde{P}_{0a}, \tilde{P}_{1a}))$, where $d_{\mathrm{TV}}$ denotes the total variation distance [56]. Thus, achieving $d_{\mathrm{TV}}(\tilde{P}_{00}, \tilde{P}_{10}) = d_{\mathrm{TV}}(\tilde{P}_{01}, \tilde{P}_{11})$ ensures equal error rates across both the groups. However, there is no closed form expression for the total variation distance between two univariate Gaussians, and KL-divergence can be thought of as a proxy using Pinsker's inequality [19]. This is similar to information theoretic argument used by Dutta et al. [30], which is why their optimization cannot guarantee outcome fairness in the way we do.

## 4 Finding The Nearest Optimal Distribution

When a given distribution $D$ is not ideal, then a natural question is to find its nearest distribution $\tilde{D}$ that is ideal. We formulate this problem as follows.

$$\underset{\tilde{D} \,:\, \tilde{D} \text{ is ideal}}{\text{minimize}} D_{\mathrm{KL}}\left(\tilde{D}||D\right).$$

In the above optimization problem, the KL-divergence objective is well-known and convex but the constraint of $D'$ being ideal is extremely non-trivial to express. We show that when the group and class-conditioned distributions $X \mid Y = i, A = a$ come from certain well-known parametric families of distributions, this constraint can be equivalently expressed as a constraint on the distribution parameters. We now give a concrete formulation of the optimization problem described in Section 3 using the constraints derived in Proposition 3.2.

$$\min_{\tilde{D}} -\frac{d}{2} + \frac{1}{2} \sum_{(i,a)} q_{ia} (\tilde{\mu}_{ia} - \mu_{ia})^T \Sigma_{ia}^{-1} (\tilde{\mu}_{ia} - \mu_{ia}) \frac{1}{2} \sum_{(i,a)} q_{ia} \left( \operatorname{tr}\left( \Sigma_{ia}^{-1} \tilde{\Sigma}_{ia} \right) - \log \det(\Sigma_{ia}^{-1} \tilde{\Sigma}_{ia}) \right)$$

$$\text{subject to } \Sigma_{ia}^{-1/2}(\mu_{ia} - \mu_{ja}) = \Sigma_{ia'}^{-1/2}(\mu_{ia'} - \mu_{ja'}), \ \Sigma_{ia}^{1/2} \Sigma_{ja}^{-1} \Sigma_{ia}^{1/2} = \Sigma_{ia'}^{1/2} \Sigma_{ja'}^{-1} \Sigma_{ia'}^{1/2} \text{ and}$$

$$\frac{q_{ia}}{q_{ja}} = \frac{q_{ia'}}{q_{ja'}}, \ \forall i, j \in \mathcal{Y}, a, a' \in \mathcal{A}.$$

For readability, we will work with binary class labels and binary group attributes in this section. However, all of our results can also be written down for multi-class and multi-group settings. In its full generality, the above problem is non-convex, and therefore, we will first propose reducing this to a convex optimization problem and then propose solving it in full generality by using line search.

## 4.1 Affirmative Action

We first focus on a class of interventions for which solving the optimization program is efficient. We define *Affirmative Action* as changing the underprivileged group to obtain the ideal distributions where fairness and accuracy are in accord.

**Theorem 4.1.** *Let $(X, Y, A)$ denote the features, binary class label, and binary group membership, respectively, of a random data point from any data distribution $D$ with $q_{ia} = \Pr(Y = i, A = a)$, for $i \in \{0, 1\}$ and $a \in \{0, 1\}$, such that $q_{10}/q_{00} = q_{11}/q_{01}$. Let $X|Y = i, A = a \sim \mathcal{N}(\mu_{ia}, \Sigma_{ia})$ be multivariate Normal distributions, with mean $\mu_{ia} \in \mathbb{R}^d$ and covariance matrix $\Sigma_{ia} \in \mathbb{R}^{d \times d}$, for $i \in \{0, 1\}$ and $a \in \{0, 1\}$. Let $\tilde{D}$ denote a distribution obtained by keeping $(Y, A)$ unchanged and only changing $X|Y = i, A = a$ to $\tilde{X}|Y = i, A = a \sim \mathcal{N}(\tilde{\mu}_{ia}, \tilde{\Sigma}_{ia})$. Then in the case of Affirmative action (changing only $\tilde{\mu}_{i0}$ and $\tilde{\Sigma}_{i0}$), we can efficiently minimize $D_{\mathrm{KL}}\left(\tilde{D}||D\right)$ as a function of the variables $\tilde{\mu}_{i0}$ and $\tilde{\Sigma}_{i0}$ subject to the constraints in Proposition 3.2, so that the Bayes optimal classifier on the optimal $\tilde{D}$ is guaranteed to be EO-fair.*

The proof for Theorem 4.1 is given in Section B of the Appendix. While we show that the optimization program is convex, obtaining a closed-form expression for the change in means and covariances is extremely cumbersome for the general case. However, we can show how the closed form expressions for $\tilde{\mu}_{i0}$ and $\tilde{\sigma}_{i0}$ look like for the univariate case in the following Corollary:

**Corollary 4.2.** *(Univariate Affirmative Action) For the case where $X|Y = i, A = a \sim \mathcal{N}(\mu_{ia}, \sigma_{ia}^2)$ are univariate normal distributions, for $i, a \in \{0, 1\}$, the optimal distribution $\tilde{D}$ from Theorem 4.1, with $\gamma^*$ being a function of the original distribution parameters, can be written down as:*

$$\tilde{\sigma}_{i0} = \gamma^* \sigma_{i1}, \quad \tilde{\mu}_{00} = \tilde{\mu}_{10} + \gamma^*(\mu_{01} - \mu_{11}), \text{ and } \tilde{\mu}_{10} = \frac{\left( q_{00} \dfrac{\mu_{00} - \gamma^*(\mu_{01} - \mu_{11})}{\sigma_{00}^2} + q_{10} \dfrac{\mu_{10}}{\sigma_{10}^2} \right)}{\left( \dfrac{q_{00}}{\sigma_{00}^2} + \dfrac{q_{10}}{\sigma_{10}^2} \right)},$$

## 4.2 Changing all Subgroups

Another intervention we can follow is to change all the subgroups of the given distribution. However, a quick check through the proof of Theorem 4.1 shows that this will lead to a non-convex program. However, just like Corollary 4.2, we can show a reasonable intervention for the univariate case, where we change all four subgroups and search over a non-convex function using line search over a fairly large grid size.

**Proposition 4.3.** *(All subgroup change for Exact Fairness) Let $(X, Y, A)$ denote the features, binary class label, and binary group membership, respectively, of a random data point from any data distribution $D$ with $q_{ia} = \Pr(Y = i, A = a)$, for $i \in \{0, 1\}$ and $a \in \{0, 1\}$, such that $q_{10}/q_{00} = q_{11}/q_{01}$, and let $X|Y = i, A = a \sim \mathcal{N}(\mu_{ia}, \sigma_{ia}^2)$ be univariate normal distributions, for $i \in \{0, 1\}$ and $a \in \{0, 1\}$. Let $\tilde{D}$ denote a distribution obtained by keeping $(Y, A)$ unchanged and only changing $X|Y = i, A = a$ to $\tilde{X}|Y = i, A = a \sim \mathcal{N}(\tilde{\mu}_{ia}, \tilde{\sigma}_{ia}^2)$. Then minimizing $D_{\mathrm{KL}}\left(\tilde{D}||D\right)$ as a function of the variables $\tilde{\mu}_{ia}$ and $\tilde{\sigma}_{ia}$ subject to the constraints in Proposition 3.2 leads to a non-convex program. Furthermore, let $\gamma^* = \arg\min_{\gamma \in (0,\infty)} \mathcal{L}_\gamma^*$ for some non-convex function of $\gamma$ that is only dependent on the original distribution parameters. Then, all the new distribution parameters $\tilde{\mu}_{ia}$ and $\tilde{\sigma}_{ia}$ can be expressed as a function of $\gamma^*$ and the original distribution parameters $\mu_{ia}$ and $\sigma_{ia}$.*

The proof of Proposition 4.3 is provided in Section B of the Appendix. We can also derive worst-case upper bounds on the error rate and the unfairness gap $\Delta_{\mathrm{EO}}$ of the Bayes optimal classifier $\tilde{h}$ on $\tilde{D}$ with respect to the original distribution $D$. These bounds show that both the accuracy loss and the fairness gap depend only on the KL divergence between $D$ and $\tilde{D}$. It also shows that the optimal value of our optimization problem can be used to approximately translate the accuracy guarantee of $\tilde{h}$ from $\tilde{D}$ to $D$.

**Proposition 4.4.** *Let $err(h, D)$ denote the error rate (expected 0-1 loss) of a classifier $h$ on the distribution $D$. Let $d_{TV}(\tilde{D}, D)$ denote the total variation distance between two distributions $\tilde{D}$ and $D$, while $D_{KL}$ denotes the KL-Divergence between them. Denote the Bayes optimal classifier on the ideal distribution $\tilde{D}$ as $\tilde{h}$ (and similarly the Bayes optimal classifier $h$. Then, we can bound the error rate and the Equal opportunity of $\tilde{h}$ on the original distribution $D$ as follows:*

$$|err(\tilde{h}, D) - err(\tilde{h}, \tilde{D})| \le \sqrt{2D_{KL}(\tilde{D}, D)} \quad and \quad \Delta_{EO}(\tilde{h}, D) \le \sqrt{8D_{\mathrm{KL}}\left(\tilde{D}||D\right)}.$$

The proof for Proposition 4.4 is given in Section B of the Appendix. The above bounds informs us that when the ideal distributions are close enough to the true distribution, we do not lose much when going towards an ideal distribution.

## 5  Case Study on Gaussian Distributions

In this section, we adapt a stylized Gaussian setting from prior work (Definition 3.1 in [59], Section 5.3 in [4]) to examine unfairness and Bayes optimal error before and after distributional interventions. We assume homoskedastic Gaussians within each group $A = a$, i.e., $\sigma_{0a} = \sigma_{1a}$, so that the Bayes classifier admits a threshold form. This enables tractable analysis of interventions aimed at achieving a fairness–accuracy trade-off. Further details are provided in Section C of the Appendix.

We evaluate four interventions: (a) the Bayes optimal classifier on the original distribution (no correction), (b) *EF Affirmative*: transforming the underprivileged group ($A = 0$) via the univariate KL program in Corollary 4.2, (c) *EF-All Subgroups*: modifying all groups to minimize KL divergence to the original distribution under exact fairness constraints (Proposition 4.3), and (d) *Mean Matching*: aligning group means while minimizing KL divergence (Proposition B.6 in Section B of the Appendix). Note that 'EF-All Subgroups' involves non-convex optimization, for which we apply line search to estimate the optimal mean–variance scaling factor $\gamma$. We report Demographic Parity ($\Delta$DP), Equal Opportunity ($\Delta$EO), KL and Jensen–Shannon divergence, as well as Bayes Error (BE) under each intervention. For the univariate Gaussian case, the Bayes-optimal thresholds are known in closed form (Lemma A.1 in Section A of the Appendix), and allow precise computation of $\Delta$DP and $\Delta$EO via standard Gaussian CDFs.

We have already demonstrated the effect of different interventions for the case of high $\Delta$EO in Figure 1. To cover another interesting case, we look at a distribution where $\Delta$DP is very high in Figure 2. Here, the affirmative action intervention transforms both the under-privileged subgroups to high-variance ones, which results in a reduction of BE and $\Delta$DP, but at the cost of a high KL/JS-divergence with respect to the true distribution. However, the EF-All intervention simply tries to match the variances of under-privileged and privileged subgroups and, as a result, achieves perfect fairness and accuracy while staying close to the true distribution. Mean Matching is very similar to EF Affirmative in this case and, as a result, has relatively high KL/JS numbers. Due to a lack of

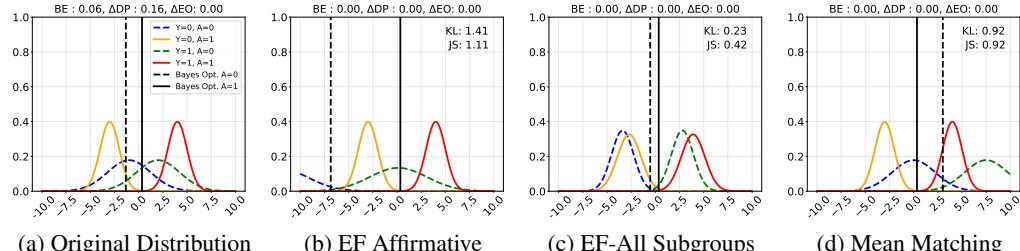

|  (a) Original Distribution | (b) EF Affirmative | (c) EF-All Subgroups | (d) Mean Matching |

Figure 2: Comparison of different interventions when the $\Delta$DP on the original distribution is high. In this case, EF-All manages to stay close to the true distribution and achieves perfect fairness and error rate, while others deviate significantly.

space, we show more results for different settings: (1) same but shifted group distributions (Figure 5), (2) the case of $\Delta$EO being close to $0$ (Figure 6), and (3) the case of a different cost sensitive risk ($t_C = 3/4$ on $\eta(x, a)$) (Figure 7), in Section C of the Appendix.

## 6    Applications: Steering the Representations of an LLM

We now empirically demonstrate how our guarantees can improve representation and generation steering in LLMs—an active area with limited theoretical grounding [37, 67, 79, 68]. Adopting the setup of Singh et al. [67], we first apply affine steering to pretrained LLM representations to reduce class-specific TPR gaps on the Bias in Bios dataset [27]. We then use the framework of Zhao et al. [79] to steer generations of movie reviews, showing that our intervention improves expressed joy for the underperforming group.

### 6.1    Reducing Per-class disparity in Multi-class Classification

In this experiment, we aim to steer data representations to reduce disparities between groups in a multi-class classification setting. We use the Bias in Bios dataset [27], which comprises web-sourced biographies labelled by profession and annotated for gender. The task is to predict the profession from the biography while ensuring fairness with respect to a chosen metric. Our experimental setup closely follows that of Singh et al. [67], who generate representations using the Llama-2 7b model [72] and propose a method called MiMiC (Mean+Covariance Matching), which steers representations via least squares alignment of the first two moments. They measure the TPR-gap (Definition 2.1), defined as: $\text{TPR-gap}_y(h) := |\mathbb{P}[h(X, A) = y \mid Y = y, A = 1] - \mathbb{P}[h(X, A) = y \mid Y = y, A = 0]|$.

They demonstrate that MiMiC significantly reduces the TPR-gap on the Llama-2 7b embeddings of the Bios dataset. We adopt the same experimental pipeline as Singh et al. [66]; further details are provided in Section D.1 of the Appendix. Our intervention, referred to as *EF Affirmative* in Figure 3, begins by estimating the first two moments of the representations from Singh et al.'s pipeline. We then apply our Affirmative Action framework to compute target moments corresponding to an ideal distribution. As in Singh et al., we estimate affine transformation parameters that map the original moments to the target ones and apply this transformation to all data representations. We use a multi-class generalization of our optimization program in Theorem 4.1 and we lay down the details of the program and intervention in Section D.1 of the Appendix.

We additionally evaluate the LEACE transformation [8], which Singh et al. [66] used as a baseline. Figure 3 reports the root-mean-square TPR-gaps across classes for various methods. Our intervention consistently reduces TPR-gaps across all classes and, in many cases, outperforms both MiMiC and LEACE. This provides empirical support for our approach: actively searching for and aligning with an ideal distribution can meaningfully reduce group disparities in representation learning.

### 6.2    Steering Activations for Joyful Generation

We steer LLM generation towards joyful movie reviews using the Gaussian Concept Steer (GCS) framework of Zhao et al. [79], which samples steering vectors from a Gaussian $v_c^l \sim \mathcal{N}(\mu_c^l, \Sigma_c^l)$ for concept $c$ at layer $l$. This improves robustness across models and datasets compared to using a single vector. Concept vectors for joy are estimated from GPT-4o [40] outputs and used to steer a Llama-3

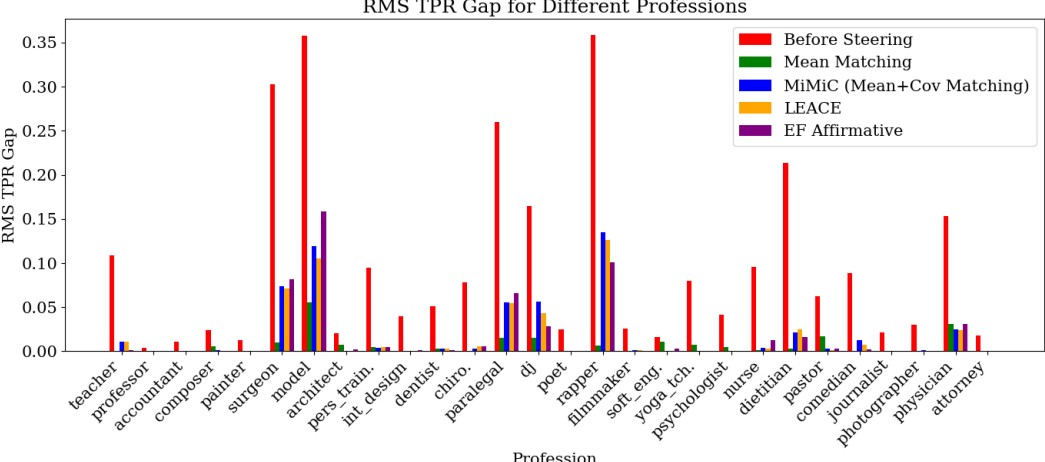

Figure 3: TPR-gap between Gender groups for all professions. All methods to steer feature representations achieve roughly the same accuracy (in the range of 0.77-0.79). Our intervention (EF Affirmative) is able to significantly reduce the TPR-gap for all professions. In many cases, it is even comparable or better than previous interventions Belrose et al. [8], Singh et al. [67].

8B model [36]. At each layer $l$, the final token representation $h^l$ is updated as $h^l = (1-a) \cdot h^l + a \cdot v_c^l$, where $a$ is the strength of the steering. Zhao et al. [79] experiment with values of $a \in (0.03, 0.08)$. We fix $a = 0.03$ for our experiments. Following [79], we evaluate the joyful tone using GPT-4 as an oracle with fixed decoding parameters. We apply the GCS framework to steer movie review generation toward joy over anger, focusing on two groups: comedy and horror reviews. For each group, we estimate Gaussian distributions of steering vectors for both directions (joyful → angry and angry → joyful), with the latter used for intervention. Full details are in Section D.2 of the Appendix.

While steering improves joyfulness overall, its effectiveness varies between groups. To address this, we apply our *EF Affirmative* intervention to nudge the joyful vector for the horror group. Let $\mathcal{N}(\mu_c^l, \Sigma_c^l)$ be the original distribution and $\mathcal{N}(\tilde{\mu}_c^l, \tilde{\Sigma}_c^l)$ be the distribution obtained after applying EF Affirmative intervention (Theorem 4.1). We define $\tilde{v}_c^l = (1 - \alpha) \cdot v_c^l + \alpha \cdot \tilde{v}_c^l$, where $v_c^l \sim \mathcal{N}(\mu_c^l, \Sigma_c^l)$ and $\tilde{v}_c^l \sim \mathcal{N}(\tilde{\mu}_c^l, \tilde{\Sigma}_c^l)$. The final steering step updates the last-token representation at layer $l$ as $h^l = (1-a) \cdot h^l + a \cdot \tilde{v}_c^l$.

Figure 4 presents the simulation results. We report the change in joyfulness (denoted as Δ-Joyful) before and after steering for both the comedy and horror review groups. For the horror group, we additionally apply our EF intervention, denoted as *Steering+EF*, to adjust the steering vectors. We vary the nudging parameter $\alpha$, where $\alpha = 0$ corresponds to the original steering vectors from Zhao et al. [79]. As expected, moderate values of $\alpha$ improve joyfulness in the horror group, but excessive nudging distorts the steering vector, reducing its effectiveness.

# 7 Discussion and Future Work

We address fair classification by introducing the notion of *ideal* distributions—those that admit the Bayes-optimal classifier satisfying exact fairness. For common parametric families, we show that identifying such distributions reduces to a KL divergence minimisation problem, subject to fairness constraints on the Bayes-optimal classifier. We characterize conditions under which this problem is feasible and tractable. Through simulations, we demonstrate that the resulting interventions can steer the original distribution towards both perfect fairness and Bayes-optimal accuracy while remaining close in distributional distance. Additionally, we show that our method extends to post-training interventions, effec-

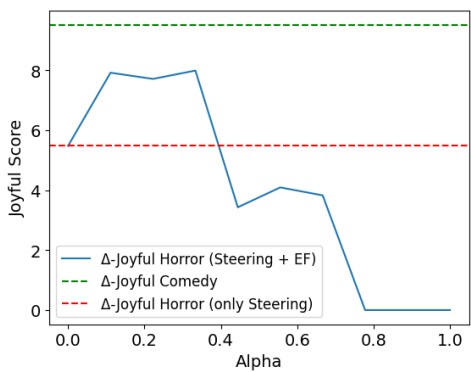

Figure 4: Change in Joyful score(Δ-Joyful) before and after adjusting the steering. Our intervention ('EF Affirmative') pushes up the effectiveness of steering the 'Horror' group, relative to the 'Comedy' group.

tively steering learned representations toward desired
behavioural outcomes.

We study the foundational problem of finding the nearest ideal distribution to a given distribution. This has many potential applications, such as correcting training data bias, learning fair representations, steering intermediate representations for fair generation, etc. Fair pre-processing or reweighing is closer to our mathematical setup, and steering intermediate representations gives a compelling application for LLMs. Our KL optimization program can be used to measure training data bias and guide data collection policies in practice, especially when the models are trained for fairness, but the inherent bias in data is unresolved.

Our theoretical results are on the fairness of the Bayes optimal classifier and the corresponding optimization programs, which are tractable and mathematically easier to analyse for parametric families of distributions, e.g., multivariate Gaussians. Multivariate Gaussians may not always fit real-world data, but can be justifiably used to model concepts and internal representations [79, 32]. There can be scenarios and applications where a parametric assumption may not fit well with the input data distribution, and hence, analyzing the Bayes optimal fair classifier will become highly non-trivial.

An important extension is to determine how to steer a given distribution within a fixed budget so that exact fairness constraints are satisfied. This is relevant when deviations from the original distribution are costly or infeasible, and can inform data collection or active learning strategies for fair classification [39, 42, 30, 3]. In such cases, it may be necessary to relax the fairness constraints and seek approximate solutions, prompting the need for efficient algorithms that operate under finite-sample settings. In a finite-sample regime, we have high probability estimates of the moments and other distribution-dependent quantities, so our approach leads to approximate fairness guarantees. To go beyond distributional assumptions, we can leverage previous work that maps given data distribution to tractable, parametric families using invertible transforms that preserve Bayes error, so our results become applicable [71]. We can also assume distributions with bounded moments instead of parametric families, and still bound the Bayes error and other relevant quantities [55]. This requires a careful analysis and is certainly a very important future direction.

The framework of ideal distributions can also be applied to audit deployed models for fairness, particularly when the evaluation data may itself be biased [2, 65, 9], offering a pathway toward more robust fairness evaluation protocols. Prior work on fairness audit has mostly focused on auditing a given model instead of auditing training or evaluation data. There is a long line of work to demonstrate that the fairness-accuracy trade-off disappears when data bias is accounted for [74, 13, 30, 51, 64, 48]. The KL gap in our formulation can be used as a metric of data bias, and Theorem 4.1 essentially gives an algorithmic recipe to find the minimally different distribution relative to the given one.

Our conditions can also be incorporated into optimization objectives to steer data generation toward ideal distributions. We demonstrate this by applying post-training interventions to steer LLM representations for multi-class fair classification and emotion control. More generally, our optimization framework and parametric conditions can be embedded into the training objectives of generative models. Modern generative approaches such as Variational Autoencoders [45] and Normalizing Flows [57] often assume parametric latent distributions, such as mixtures of Gaussians. Embedding fairness-aware ideality conditions into these objectives enables the generation of fair and accurate data distributions that remain close to the original. This represents a promising direction for fair generative modeling, an area of growing theoretical and practical interest [76, 23, 5, 73, 70, 69].

**Broader Impact**: Our work improves theoretical understanding of data bias and proposes an optimization program for steering data distributions towards provably exact fairness guarantees. The real-world societal biases in data and the fairness harms are complex and more nuanced, where our fairness interventions can provide good guiding principles but no silver bullet to solve the real-world problems.

**Acknowledgments**: Mohit Sharma would like to thank Prof. Masashi Sugiyama and the Imperfect Information Team at RIKEN-AIP for hosting him while the author was submitting and rebutting this paper. The authors would also like to thank Prof. Manuj Mukherjee at IIIT Delhi for initial discussions around Dutta et al. [30] and the optimization programs. Rajiv Ratn Shah is partly supported by the Infosys Center for AI, the Center of Design and New Media, and the Center of Excellence in Healthcare at IIIT Delhi.

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

## A   Proofs for Section 3: Ideal Distributions for Fair classification

We will require a helper result about threshold classifiers to prove our next set of results.

**Lemma A.1.** *Let $\eta(x, a) = \Pr(Y = 1|X = x, A = a)$, $q_{ia} = \Pr(Y = i, A = a)$ and $p_{ia}(x) = \Pr(X = x \mid Y = i, A = a)$. Then the Bayes optimal classifier can be written as $h^*(x, a) = \mathbb{I}\left(\log \frac{p_{1a}(x)}{p_{0a}(x)} \geq \log \frac{q_{0a}}{q_{1a}}\right)$.*

*Proof.* Let $\eta(x, a) = \Pr(Y = 1|X = x, A = a)$, $q_{ia} = \Pr(Y = i, A = a)$ and $p_{ia}(x) = \Pr(X = x \mid Y = i, A = a)$. We consider group-aware threshold classifiers on $D$ of the form $h_t(x, a) = \mathbb{I}(\eta(x, a) \geq t)$, which can be equivalently written as

$$
\begin{aligned}
h_t(x, a) &= \mathbb{I}(\eta(x, a) \geq t) \\
&= \mathbb{I}\left(\Pr(Y = 1 \mid X = x, A = a) \geq t\right) \\
&= \mathbb{I}\left(\frac{\Pr(Y = 1 \mid X = x, A = a)}{\Pr(Y = 0 \mid X = x, A = a)} \geq \frac{t}{1 - t}\right) \\
&= \mathbb{I}\left(\frac{\Pr(Y = 1, X = x, A = a)}{\Pr(Y = 0, X = x, A = a)} \geq \frac{t}{1 - t}\right) \\
&= \mathbb{I}\left(\frac{\Pr(X = x \mid Y = 1, A = a)\Pr(Y = 1, A = a)}{\Pr(X = x \mid Y = 0, A = a)\Pr(Y = 0, A = a)} \geq \frac{t}{1 - t}\right) \\
&= \mathbb{I}\left(\frac{p_{1a}(x)}{p_{0a}(x)} \geq \frac{t}{1 - t} \cdot \frac{q_{0a}}{q_{1a}}\right) \\
&= \mathbb{I}\left(\log \frac{p_{1a}(x)}{p_{0a}(x)} \geq \log \frac{t}{1 - t} + \log \frac{q_{0a}}{q_{1a}}\right).
\end{aligned}
$$

It is well-known that the group-aware Bayes optimal classifier $h^* = h_{1/2}$ by setting $t = 1/2$, or equivalently,

$$
h^*(x, a) = h_{1/2}(x, a) = \mathbb{I}\left(\log \frac{p_{1a}(x)}{p_{0a}(x)} \geq \log \frac{q_{0a}}{q_{1a}}\right).
$$

$\square$

We now prove Proposition 3.2 from the main paper.

**Proposition A.2.** *(Proposition 3.2 in the main text) Let $(X, Y, A)$ denote the features, class label, and group membership, respectively, of a random data point from any data distribution $D$ with $q_{ia} = \Pr(Y = i, A = a)$, for $i \in \mathcal{Y}$ and $a \in \mathcal{A}$. Let $X|Y = i, A = a \sim \mathcal{N}(\mu_{ia}, \Sigma_{ia})$ be multivariate Normal distributions with mean $\mu_{ia} \in \mathbb{R}^d$ and covariance matrix $\Sigma_{ia} \in \mathbb{R}^{d \times d}$, for $i \in \mathcal{Y}$ and $a \in \mathcal{A}$. If the means $\mu_{ia}$ and the covariance matrices $\Sigma_{ia}$ satisfy*

$$
\begin{aligned}
&\Sigma_{ia}^{-1/2}(\mu_{ia} - \mu_{ja}) = \Sigma_{ia'}^{-1/2}(\mu_{ia'} - \mu_{ja'}) \quad \text{and} \\
&\Sigma_{ia}^{1/2}\Sigma_{ja}^{-1}\Sigma_{ia}^{1/2} = \Sigma_{ia'}^{1/2}\Sigma_{ja'}^{-1}\Sigma_{ia'}^{1/2} \quad \text{and} \quad \frac{q_{ia}}{q_{ja}} = \frac{q_{ia'}}{q_{ja'}}, \forall i, j \in \mathcal{Y}, a, a' \in \mathcal{A},
\end{aligned}
$$

*then the group-aware Bayes optimal classifier on $D$ satisfies equal opportunity.*

*Proof.* The Bayes optimal classifier for group $A = a$ in a multi-class setting can be written down as a maximum over posterior probabilities:

$$h^*(x, a) = \arg\max_{y \in \mathcal{Y}} \eta_y(x, a), \text{ where } \eta_y(x, a) = \Pr\left(Y = y | X = x, A = a\right).$$

We can say that $h^*(x, a) = y$, whenever the following happens:

$$h^*(x, a) = \arg\max_{y \in \mathcal{Y}} \eta_y(x, a) = \mathbb{I}\left(\frac{\eta_y(x, a)}{\eta_i(x, a)} \geq 1, \forall i \in \mathcal{Y}\right) = \mathbb{I}\left(\log\frac{p_{ya}(X)}{p_{ia}(X)} \geq \log\frac{q_{ia}}{q_{ya}}, \forall i \in \mathcal{Y}\right)$$

Using the above simplification, the EO-fairness condition $\Pr\left(h^*(X, A) = y \mid Y = y, A = a\right) = \Pr\left(h^*(X, A) = y \mid Y = 1, A = a'\right) \ \forall y \in \mathcal{Y}, a, a' \in \mathcal{A}$ means

$$\Pr\left(\log\frac{p_{ya}(X)}{p_{ia}(X)} \geq \log\frac{q_{ia}}{q_{ya}}, \forall i \in \mathcal{Y} \mid Y = y, A = a\right)$$

$$= \Pr\left(\log\frac{p_{ya'}(X)}{p_{ia'}(X)} \geq \log\frac{q_{ia'}}{q_{ya'}}, \forall i \in \mathcal{Y} \mid Y = y, A = a'\right).$$

Since $X|Y = i, A = a \sim \mathcal{N}(\mu_{ia}, \Sigma_{ia})$ are multivariate Normal distributions, their probability densities are

$$p_{ia}(x) = (2\pi)^{-d/2} \det(\Sigma_{ia})^{-1/2} \exp\left(-\frac{1}{2}(x - \mu_{ia})^T \Sigma_{ia}^{-1}(x - \mu_{ia})\right).$$

Now we can write

$$\log\frac{p_{ya}(x)}{p_{ia}(x)}$$

$$= \frac{1}{2}\left((x - \mu_{ia})^T \Sigma_{ia}^{-1}(x - \mu_{ia}) - (x - \mu_{ya})^T \Sigma_{ya}^{-1}(x - \mu_{ya}) + \log\det(\Sigma_{ia}) - \log\det(\Sigma_{ya})\right)$$

$$= \frac{1}{2}\left((\Sigma_{ya}^{1/2}r + \mu_{ya} - \mu_{ia})^T \Sigma_{ia}^{-1}(\Sigma_{ya}^{1/2}r + \mu_{ya} - \mu_{ia}) - r^T r - \log\det(\Sigma_{ya}^{1/2}\Sigma_{ia}^{-1}\Sigma_{ya}^{1/2})\right)$$

$$\text{by substituting } x = \Sigma_{ya}^{1/2}r + \mu_{ya}, \text{ where } r \sim \mathcal{N}(0, I_{d\times d})$$

$$= \frac{1}{2} r^T \Sigma_{ya}^{1/2}\Sigma_{ia}^{-1}\Sigma_{ya}^{1/2}r + (\mu_{ya} - \mu_{ia})^T \Sigma_{ia}^{-1}\Sigma_{ya}^{1/2}r + \frac{1}{2}(\mu_{ya} - \mu_{ia})^T \Sigma_{ia}^{-1}(\mu_{ya} - \mu_{ia})$$

$$- \frac{1}{2} r^T r - \frac{1}{2}\log\det(\Sigma_{ya}^{1/2}\Sigma_{ia}^{-1}\Sigma_{ya}^{1/2})$$

Let us denote the above expression as $E_{yi}(r)$. We can now write the group TPR as:

$$\Pr\left(\log\frac{p_{ya}(X)}{p_{ia}(X)} \geq \log\frac{q_{ia}}{q_{ya}}, \forall i \in \mathcal{Y} \mid Y = y, A = a\right) = \Pr\left(E_{yi}(R) \geq \log\frac{q_{ia}}{q_{ya}}, \forall i \in \mathcal{Y}\right),$$

for $R \sim \mathcal{N}(\bar{0}, I_{d\times d})$. Now if we have $\frac{q_{ya}}{q_{ia}} = \frac{q_{ya'}}{q_{ia'}}$ and

$$\Sigma_{ya}^{-1/2}(\mu_{ya} - \mu_{ia}) = \Sigma_{ya'}^{-1/2}(\mu_{ya'} - \mu_{ia'}) \quad \text{and} \quad \Sigma_{ya}^{1/2}\Sigma_{ia}^{-1}\Sigma_{ya}^{1/2} = \Sigma_{ya'}^{1/2}\Sigma_{ia'}^{-1}\Sigma_{ya'}^{1/2},$$

then the probability of the above event written in terms $R \sim \mathcal{N}(\bar{0}, I_{d\times d})$ becomes identical $\forall a, a' \in \mathcal{A}, \ i \in \mathcal{Y}$. Hence, the Bayes optimal classifier satisfies equal opportunity with these set of conditions.

$\square$

**Proposition A.3.** *(Proposition 3.3 in the main text) Let $(X, Y, A)$ denote the features, binary class label, and binary group membership, respectively, of a random data point from any data distribution $D$ with $q_{ia} = \Pr(Y = i, A = a)$, for $i \in \{0, 1\}$ and $a \in \{0, 1\}$, and let $X|Y = i, A = a \sim \mathcal{N}(\mu_{ia}, \sigma_{ia}^2)$ be univariate normal distributions, for $i \in \{0, 1\}$ and $a \in \{0, 1\}$. Then the distribution $D$ is* ideal *for equal opportunity (see Definition 3.1) if and only if*

$$\frac{\mu_{01} - \mu_{11}}{\sigma_{11}} = \frac{\mu_{00} - \mu_{10}}{\sigma_{10}}, \quad \frac{\sigma_{11}}{\sigma_{01}} = \frac{\sigma_{10}}{\sigma_{00}}, \quad \frac{q_{10}}{q_{00}} = \frac{q_{11}}{q_{01}}.$$

*Proof.* For any cost matrix $C \in \mathbb{R}^{2\times2}$, the group-aware classifier that minimizes its corresponding cost-sensitive risk is given by $\mathbb{I}\left(\eta(x,a) \geq t_C\right)$, for a threshold $t_C = (c_{10} - c_{00})/(c_{10} - c_{00} + c_{01} - c_{11}) \in [0,1]$; see Equation (2) in [33] and [63]. The distribution $D$ is *ideal* for equal opportunity if $\Pr\left(\eta(X,A) \geq t \mid Y = i, A = 0\right) = \Pr\left(\eta(X,A) \geq t \mid Y = i, A = 1\right)$, for all thresholds $t \in [0,1]$ and $i \in \{0,1\}$. Since the CDFs are identical, the random variables $\eta(X,A) \mid Y = i, A = 0$ and $\eta(X,A) \mid Y = i, A = 1$ must be identical. Note that

$$
\begin{aligned}
\eta(x,a) &= \Pr\left(Y = 1 \mid X = x, A = a\right) \\
&= \frac{\Pr\left(Y = 1, X = x, A = a\right)}{\sum_{i=0}^{1} \Pr\left(Y = i, X = x, A = a\right)} \\
&= \frac{\Pr\left(Y = 1, A = a\right) \Pr\left(X = x \mid Y = 1, A = a\right)}{\sum_{i=0}^{1} \Pr\left(Y = i, A = a\right) \Pr\left(X = x \mid Y = i, A = a\right)} \\
&= \frac{q_{1a} P_{1a}(x)}{\sum_{i=0}^{1} q_{ia} P_{ia}(x)} \\
&= \frac{q_{1a}\sigma_{1a}^{-1} \exp\left(-\dfrac{(x - \mu_{1a})^2}{2\sigma_{1a}^2}\right)}{\sum_{i=0}^{1} q_{ia}\sigma_{ia}^{-1} \exp\left(-\dfrac{(x - \mu_{ia})^2}{2\sigma_{ia}^2}\right)} \\
&= \frac{1}{1 + \exp\left(\dfrac{(x - \mu_{1a})^2}{2\sigma_{1a}^2} - \dfrac{(x - \mu_{0a})^2}{2\sigma_{0a}^2} + \log\dfrac{q_{0a}\sigma_{1a}}{q_{1a}\sigma_{0a}}\right)} \\
&= \frac{1}{1 + \exp\left(\dfrac{(\mu_{ia} + r\sigma_{ia} - \mu_{1a})^2}{2\sigma_{1a}^2} - \dfrac{(\mu_{ia} + r\sigma_{ia} - \mu_{0a})^2}{2\sigma_{0a}^2} + \log\dfrac{q_{0a}\sigma_{1a}}{q_{1a}\sigma_{0a}}\right)} \\
&= \begin{cases}
\dfrac{1}{1 + \exp\left(\dfrac{1}{2}\left(\dfrac{\sigma_{0a}^2}{\sigma_{1a}^2} - 1\right) r^2 - \dfrac{\sigma_{0a}(\mu_{1a} - \mu_{0a})}{\sigma_{1a}^2}r + \dfrac{(\mu_{1a} - \mu_{0a})^2}{2\sigma_{1a}^2} + \log\dfrac{q_{0a}\sigma_{1a}}{q_{1a}\sigma_{0a}}\right)}, & \text{for } i = 0 \\[2em]
\dfrac{1}{1 + \exp\left(\dfrac{1}{2}\left(1 - \dfrac{\sigma_{1a}^2}{\sigma_{0a}^2}\right) r^2 - \dfrac{\sigma_{1a}(\mu_{0a} - \mu_{1a})}{\sigma_{0a}^2}r + \dfrac{(\mu_{0a} - \mu_{1a})^2}{2\sigma_{0a}^2} + \log\dfrac{q_{0a}\sigma_{1a}}{q_{1a}\sigma_{0a}}\right)}, & \text{for } i = 1.
\end{cases}
\end{aligned}
$$

If $X|Y = i, A = a \sim \mathcal{N}(\mu_{ia}, \sigma_{ia}^2)$, then $X \mid Y = i, A = a \sim \mathcal{N}(\mu_{ia}, \sigma_{ia}^2)$. Thus, for $\eta(X,A) \mid Y = i, A = 0$ and $\eta(X,A) \mid Y = i, A = 1$ to be identical, we must have

$$
\frac{1}{2}\left(\frac{\sigma_{00}^2}{\sigma_{10}^2} - 1\right) R^2 - \frac{\sigma_{00}(\mu_{10} - \mu_{00})}{\sigma_{10}^2}R + \frac{(\mu_{10} - \mu_{00})^2}{2\sigma_{10}^2} + \log\frac{q_{00}\sigma_{10}}{q_{10}\sigma_{00}} \quad \text{and}
$$

$$
\frac{1}{2}\left(\frac{\sigma_{01}^2}{\sigma_{11}^2} - 1\right) R^2 - \frac{\sigma_{01}(\mu_{11} - \mu_{01})}{\sigma_{11}^2}R + \frac{(\mu_{11} - \mu_{01})^2}{2\sigma_{11}^2} + \log\frac{q_{01}\sigma_{11}}{q_{11}\sigma_{01}}
$$

as identically distributed for $R \sim \mathcal{N}(0,1)$. Similarly, we must also have

$$
\frac{1}{2}\left(1 - \frac{\sigma_{10}^2}{\sigma_{00}^2}\right) R^2 - \frac{\sigma_{10}(\mu_{00} - \mu_{10})}{\sigma_{00}^2}R + \frac{(\mu_{00} - \mu_{10})^2}{2\sigma_{00}^2} + \log\frac{q_{00}\sigma_{10}}{q_{10}\sigma_{00}} \quad \text{and}
$$

$$
\frac{1}{2}\left(1 - \frac{\sigma_{11}^2}{\sigma_{01}^2}\right) R^2 - \frac{\sigma_{11}(\mu_{01} - \mu_{11})}{\sigma_{01}^2}R + \frac{(\mu_{01} - \mu_{11})^2}{2\sigma_{01}^2} + \log\frac{q_{01}\sigma_{11}}{q_{11}\sigma_{01}}
$$

as identically distributed for $R \sim \mathcal{N}(0,1)$. Therefore, we must have

$$
\frac{\mu_{01} - \mu_{11}}{\sigma_{11}} = \frac{\mu_{00} - \mu_{10}}{\sigma_{10}} \quad \text{and} \quad \frac{\sigma_{11}}{\sigma_{01}} = \frac{\sigma_{10}}{\sigma_{00}} \quad \text{and} \quad \frac{q_{10}}{q_{00}} = \frac{q_{11}}{q_{01}}.
$$

In the other direction, it is easier to prove that the above conditions imply the distribution to be ideal. It can be proved by simply backtracking the steps above. $\qquad\square$

# B    Proofs for Section 4

We first derive the KL divergence between two distributions, where each subgroup in the distribution follows a multivariate normal distribution.

**Lemma B.1.** *Let $(X, Y, A)$ denote the features, binary class label, and binary group membership, respectively, of a random data point from any data distribution $D$ with $q_{ia} = \Pr(Y = i, A = a)$, for $i \in \mathcal{Y}$ and $a \in \mathcal{A}$. Let $X | Y = i, A = a \sim \mathcal{N}(\mu_{ia}, \Sigma_{ia})$ be multivariate Normal distributions with mean $\mu_{ia} \in \mathbb{R}^d$ and covariance matrix $\Sigma_{ia} \in \mathbb{R}^{d \times d}$. Let $\tilde{D}$ denote a distribution obtained by keeping $(Y, A)$ unchanged and only changing $X | Y = i, A = a$ to $\tilde{X} | Y = i, A = a \sim \mathcal{N}(\tilde{\mu}_{ia}, \tilde{\Sigma}_{ia})$. Then,*

$$D_{\mathrm{KL}}\left(\tilde{D} \| D\right) = -\frac{d}{2} + \frac{1}{2} \sum_{(i,a)} q_{ia} (\tilde{\mu}_{ia} - \mu_{ia})^T \Sigma_{ia}^{-1} (\tilde{\mu}_{ia} - \mu_{ia})$$

$$+ \frac{1}{2} \sum_{(i,a)} q_{ia} \left( \mathrm{tr}\left(\Sigma_{ia}^{-1} \tilde{\Sigma}_{ia}\right) - \log \det(\Sigma_{ia}^{-1} \tilde{\Sigma}_{ia}) \right).$$

*Proof.*

$$D_{\mathrm{KL}}\left(\tilde{D} \| D\right)$$

$$= \sum_{(x,i,a)} \Pr\left(\tilde{X} = x, \tilde{Y} = i, \tilde{A} = a\right) \log \frac{\Pr\left(\tilde{X} = x, \tilde{Y} = i, \tilde{A} = a\right)}{\Pr\left(X = x, Y = i, A = a\right)}$$

$$= \sum_{(x,i,a)} \Pr\left(\tilde{Y} = i, \tilde{A} = a\right) \Pr\left(\tilde{X} = x \mid \tilde{Y} = i, \tilde{A} = a\right)$$

$$\log \frac{\Pr\left(\tilde{Y} = i, \tilde{A} = a\right) \Pr\left(\tilde{X} = x \mid \tilde{Y} = i, \tilde{A} = a\right)}{\Pr\left(Y = y, A = a\right) \Pr\left(X = x \mid Y = i, A = a\right)}$$

$$= \sum_{(x,i,a)} \Pr\left(Y = i, A = a\right) \Pr\left(\tilde{X} = x \mid Y = i, A = a\right)$$

$$\log \frac{\Pr\left(Y = i, A = a\right) \Pr\left(\tilde{X} = x \mid Y = i, A = a\right)}{\Pr\left(Y = i, A = a\right) \Pr\left(X = x \mid Y = i, A = a\right)}$$

$$= \sum_{(i,a)} q_{ia} \sum_x \Pr\left(\tilde{X} = x \mid Y = i, A = a\right) \log \frac{\Pr\left(\tilde{X} = x \mid Y = i, A = a\right)}{\Pr\left(X = x \mid Y = i, A = a\right)}$$

$$= \sum_{(i,a)} q_{ia} D_{\mathrm{KL}}\left(\tilde{P}_{ia} \| P_{ia}\right)$$

$P_{ia}$ denotes the distribution of $X \mid Y = i, A = a \sim \mathcal{N}(\mu_{ia}, \Sigma_{ia})$ and $\tilde{P}_{ia}$ denotes the distribution of $\tilde{X} \mid Y = i, A = a \sim \mathcal{N}(\tilde{\mu}_{ia}, \tilde{\Sigma}_{ia})$. Their probability densities are

$$p_{ia}(x) = (2\pi)^{-d/2} \det(\Sigma_{ia})^{-1/2} \exp\left(-\frac{1}{2}(x - \mu_{ia})^T \Sigma_{ia}^{-1} (x - \mu_{ia})\right) \quad \text{and}$$

$$\tilde{p}_{ia}(x) = (2\pi)^{-d/2} \det(\tilde{\Sigma}_{ia})^{-1/2} \exp\left(-\frac{1}{2}(x - \tilde{\mu}_{ia})^T \tilde{\Sigma}_{ia}^{-1} (x - \tilde{\mu}_{ia})\right),$$

respectively. Hence, the Kullback-Leibler divergence between $\tilde{P}_{ia}$ and $P_{ia}$ can be written as

$$D_{\mathrm{KL}}\left(\tilde{P}_{ia}||P_{ia}\right)$$

$$= \mathrm{E}\left[\log\frac{\tilde{p}_{ia}(\tilde{X})}{p_{ia}(\tilde{X})} \mid Y = i, A = a\right]$$

$$= \frac{1}{2}\mathrm{E}\Big[(\tilde{X} - \mu_{ia})^T\Sigma_{ia}^{-1}(\tilde{X} - \mu_{ia}) - (\tilde{X} - \tilde{\mu}_{ia})^T\tilde{\Sigma}_{ia}^{-1}(\tilde{X} - \tilde{\mu}_{ia})$$

$$- \log\det(\Sigma_{ia}^{1/2}\tilde{\Sigma}_{ia}^{-1}\Sigma_{ia}^{1/2}) \mid Y = i, A = a\Big]$$

$$= \frac{1}{2}\mathrm{E}\left[(\tilde{X} - \mu_{ia})^T\Sigma_{ia}^{-1}(\tilde{X} - \mu_{ia}) \mid Y = i, A = a\right] - \frac{d}{2} - \frac{1}{2}\log\det(\Sigma_{ia}^{1/2}\tilde{\Sigma}_{ia}^{-1}\Sigma_{ia}^{1/2})$$

$$\text{using } \mathrm{E}\left[(\tilde{X} - \tilde{\mu}_{ia})^T\tilde{\Sigma}_{ia}^{-1}(\tilde{X} - \tilde{\mu}_{ia}) \mid Y = i, A = a\right] = \tilde{\Sigma}_{ia}^{-1} \bullet \tilde{\Sigma}_{ia} = \mathrm{tr}\left(I_{d\times d}\right) = d$$

$$= \frac{1}{2}\mathrm{E}\left[(\tilde{X} - \tilde{\mu}_{ia} + \tilde{\mu}_{ia} - \mu_{ia})^T\Sigma_{ia}^{-1}(\tilde{X} - \tilde{\mu}_{ia} + \tilde{\mu}_{ia} - \mu_{ia}) \mid Y = i, A = a\right]$$

$$- \frac{d}{2} + \frac{1}{2}\log\det(\Sigma_{ia}^{1/2}\tilde{\Sigma}_{ia}^{-1}\Sigma_{ia}^{1/2})$$

$$= \frac{1}{2}\mathrm{tr}\left(\Sigma_{ia}^{-1}\tilde{\Sigma}_{ia}\right) + \frac{1}{2}(\tilde{\mu}_{ia} - \mu_{ia})^T\Sigma_{ia}^{-1}(\tilde{\mu}_{ia} - \mu_{ia}) - \frac{d}{2} + \frac{1}{2}\log\det(\Sigma_{ia}^{1/2}\tilde{\Sigma}_{ia}^{-1}\Sigma_{ia}^{1/2}).$$

The Kullback-Leibler divergence between $\tilde{D}$ and $D$ can now be written as

$$D_{\mathrm{KL}}\left(\tilde{D}||D\right) = \sum_{(i,a)} q_{ia}D_{\mathrm{KL}}\left(\tilde{P}_{ia}||P_{ia}\right)$$

$$= \sum_{(i,a)} q_{ia}\left(\frac{1}{2}\mathrm{tr}\left(\Sigma_{ia}^{-1}\tilde{\Sigma}_{ia}\right) + \frac{1}{2}(\tilde{\mu}_{ia} - \mu_{ia})^T\Sigma_{ia}^{-1}(\tilde{\mu}_{ia} - \mu_{ia}) - \frac{d}{2} + \frac{1}{2}\log\det(\Sigma_{ia}^{1/2}\tilde{\Sigma}_{ia}^{-1}\Sigma_{ia}^{1/2})\right)$$

$$= -\frac{d}{2} + \frac{1}{2}\sum_{(i,a)} q_{ia}\left(\mathrm{tr}\left(\Sigma_{ia}^{-1}\tilde{\Sigma}_{ia}\right) + (\tilde{\mu}_{ia} - \mu_{ia})^T\Sigma_{ia}^{-1}(\tilde{\mu}_{ia} - \mu_{ia}) + \log\det(\Sigma_{ia}^{1/2}\tilde{\Sigma}_{ia}^{-1}\Sigma_{ia}^{1/2})\right)$$

$$= -\frac{d}{2} + \frac{1}{2}\sum_{(i,a)} q_{ia}(\tilde{\mu}_{ia} - \mu_{ia})^T\Sigma_{ia}^{-1}(\tilde{\mu}_{ia} - \mu_{ia}) + \frac{1}{2}\sum_{(i,a)} q_{ia}\left(\mathrm{tr}\left(\Sigma_{ia}^{-1}\tilde{\Sigma}_{ia}\right) + \log\det(\Sigma_{ia}\tilde{\Sigma}_{ia}^{-1})\right)$$

$$= -\frac{d}{2} + \frac{1}{2}\sum_{(i,a)} q_{ia}(\tilde{\mu}_{ia} - \mu_{ia})^T\Sigma_{ia}^{-1}(\tilde{\mu}_{ia} - \mu_{ia}) + \frac{1}{2}\sum_{(i,a)} q_{ia}\left(\mathrm{tr}\left(\Sigma_{ia}^{-1}\tilde{\Sigma}_{ia}\right) - \log\det(\Sigma_{ia}^{-1}\tilde{\Sigma}_{ia})\right)$$

$$\square$$

**Theorem B.2.** *(Theorem 4.1 in the main text) Let $(X, Y, A)$ denote the features, binary class label, and binary group membership, respectively, of a random data point from any data distribution $D$ with $q_{ia} = \Pr\left(Y = i, A = a\right)$, for $i \in \{0, 1\}$ and $a \in \{0, 1\}$, such that $q_{10}/q_{00} = q_{11}/q_{01}$. Let $X|Y = i, A = a \sim \mathcal{N}(\mu_{ia}, \Sigma_{ia})$ be multivariate Normal distributions, with mean $\mu_{ia} \in \mathbb{R}^d$ and covariance matrix $\Sigma_{ia} \in \mathbb{R}^{d\times d}$, for $i \in \{0, 1\}$ and $a \in \{0, 1\}$. Let $\tilde{D}$ denote a distribution obtained by keeping $(Y, A)$ unchanged and only changing $X|Y = i, A = a$ to $\tilde{X}|Y = i, A = a \sim \mathcal{N}(\tilde{\mu}_{ia}, \tilde{\Sigma}_{ia})$. Then in the case of Affirmative action (changing only $\tilde{\mu}_{i0}$ and $\tilde{\Sigma}_{i0}$), we can efficiently minimize $D_{\mathrm{KL}}\left(\tilde{D}||D\right)$ as a function of the variables $\tilde{\mu}_{i0}$ and $\tilde{\Sigma}_{i0}$ subject to the constraints in Proposition 1.2, so that the Bayes optimal classifier on the optimal $\tilde{D}$ is guaranteed to be EO-fair.*

*Proof.* Using Lemma B.1 and Proposition 1.2, our objective is to minimize

$$D_{\mathrm{KL}}\left(\tilde{D}||D\right)$$

$$= -\frac{d}{2} + \frac{1}{2}\sum_{(i,a)} q_{ia}(\tilde{\mu}_{ia} - \mu_{ia})^T\Sigma_{ia}^{-1}(\tilde{\mu}_{ia} - \mu_{ia}) + \frac{1}{2}\sum_{(i,a)} q_{ia}\left(\mathrm{tr}\left(\Sigma_{ia}^{-1}\tilde{\Sigma}_{ia}\right) - \log\det(\Sigma_{ia}^{-1}\tilde{\Sigma}_{ia})\right),$$

subject to the constraints

$$\tilde{\Sigma}_{10}^{-1/2}(\tilde{\mu}_{10} - \tilde{\mu}_{00}) = \tilde{\Sigma}_{11}^{-1/2}(\tilde{\mu}_{11} - \tilde{\mu}_{01}) \quad \text{and} \quad \tilde{\Sigma}_{10}^{1/2}\tilde{\Sigma}_{00}^{-1}\tilde{\Sigma}_{10}^{1/2} = \tilde{\Sigma}_{11}^{1/2}\tilde{\Sigma}_{01}^{-1}\tilde{\Sigma}_{11}^{1/2}.$$

Suppose $\tilde{\Sigma}_{i0}$ and $\tilde{\Sigma}_{i1}$ do not commute. The constraints can be equivalently rewritten as follows.

$$\tilde{\mu}_{10} - \tilde{\mu}_{00} = \tilde{\Sigma}_{10}^{1/2}\tilde{\Sigma}_{11}^{-1/2}(\tilde{\mu}_{11} - \tilde{\mu}_{01}) \quad \text{and} \quad \tilde{\Sigma}_{11}^{-1/2}\tilde{\Sigma}_{10}^{1/2}\tilde{\Sigma}_{00}^{-1}\tilde{\Sigma}_{10}^{1/2}\tilde{\Sigma}_{11}^{-1/2} = \tilde{\Sigma}_{01}^{-1}.$$

Let $\Gamma = \tilde{\Sigma}_{i0}^{1/2}\tilde{\Sigma}_{i1}^{-1/2}$. For any fixed positive semidefinite matrix $\Gamma \in \mathbb{R}^{d \times d}$, our optimization problem can be divided into two separate parts that minimize

$$\sum_{(i,a)} q_{ia}(\tilde{\mu}_{ia} - \mu_{ia})^T \Sigma_{ia}^{-1}(\tilde{\mu}_{ia} - \mu_{ia}) \quad \text{subject to} \quad \tilde{\mu}_{10} - \tilde{\mu}_{00} = \Gamma(\tilde{\mu}_{11} - \tilde{\mu}_{01})$$

over $\tilde{\mu}_{ia} \in \mathbb{R}^d$, for $i, a \in \{0, 1\}$, and minimize (after substituting $\tilde{\Sigma}_{i0}^{1/2} = \Gamma\tilde{\Sigma}_{i1}^{1/2}$)

$$\sum_{i=0}^{1} q_{i0}\left(\text{tr}\left(\Sigma_{i0}^{-1}\left(\Gamma\tilde{\Sigma}_{i0}^{1/2}\right)^2\right) - \log\det(\Sigma_{i0}^{-1}\left(\Gamma\tilde{\Sigma}_{i0}^{1/2}\right)^2)\right) + q_{i1}\left(\text{tr}\left(\Sigma_{i1}^{-1}\tilde{\Sigma}_{i1}\right) - \log\det(\Sigma_{i1}^{-1}\tilde{\Sigma}_{i1})\right),$$

subject to $\Gamma\tilde{\Sigma}_{11}^{1/2}\tilde{\Sigma}_{00}^{-1}\Gamma = \tilde{\Sigma}_{11}^{1/2}\tilde{\Sigma}_{01}^{-1}$

over symmetric, positive semidefinite matrix-valued variable $\tilde{\Sigma}_{i1} \in \mathbb{R}^{d \times d}$, for $i \in \{0, 1\}$. The first optimization in $\tilde{\mu}_{ia}$ is a constrained eigenvalue problem with linear constraints, i.e., minimize $x^T A x + x^T b$ subject to $x^T c = e$ [35].

Let's consider the case of *Affirmative Action*, where we only change the means $\tilde{\mu}_{i0}$ and the covariance matrices $\tilde{\Sigma}_{i0}$ for the underprivileged group but keep those for the privileged group unchanged, i.e., $\tilde{\mu}_{i1} = \mu_{i1}$ and $\tilde{\Sigma}_{i1} = \Sigma_{i1}$. In that case, $\tilde{\Sigma}_{00}^{1/2} = \Gamma\Sigma_{01}^{1/2}$ and $\tilde{\Sigma}_{10}^{1/2} = \Gamma\Sigma_{11}^{1/2}$ get fixed. By substituting $\tilde{\mu}_{10} = \tilde{\mu}_{00} + \Gamma(\tilde{\mu}_{11} - \tilde{\mu}_{01}) = \tilde{\mu}_{00} + \Gamma(\mu_{11} - \mu_{01})$, we only need to optimize

$$q_{00}(\tilde{\mu}_{00} - \mu_{00})^T\Sigma_{00}^{-1}(\tilde{\mu}_{00} - \mu_{00}) + q_{10}(\tilde{\mu}_{00} + \Gamma(\mu_{11} - \mu_{01}) - \mu_{10})^T\Sigma_{10}^{-1}(\tilde{\mu}_{00} + \Gamma(\mu_{11} - \mu_{01}) - \mu_{10}),$$

or equivalently (ignoring the terms independent of $\tilde{\mu}_{00}$),

$$\tilde{\mu}_{00}^T\left(q_{00}\Sigma_{00}^{-1} + q_{10}\Sigma_{10}^{-1}\right)\tilde{\mu}_{00} - 2\left(\Sigma_{00}^{-1}\mu_{00} + \Sigma_{10}^{-1}\mu_{10} - \Sigma_{10}^{-1}\Gamma(\mu_{11} - \mu_{01})\right)^T\tilde{\mu}_{00}.$$

This is a convex objective in $\tilde{\mu}_{00}$ because its Hessian is positive semidefinite, i.e., $q_{00}\Sigma_{00}^{-1} + q_{10}\Sigma_{10}^{-1} \succeq 0$ [15]. By equating the gradient to zero, we get the optimal solution for $\tilde{\mu}_{00}$, and we denote it by $\mu_{00}^*(\Gamma)$. Thus, the optimal solutions $\mu_{00}^*(\Gamma), \mu_{10}^*(\Gamma), \Sigma_{00}^*(\Gamma), \Sigma_{10}^*(\Gamma)$ for a fixed positive semidefinite $\Gamma \in \mathbb{R}^{d \times d}$ are given by

$$\mu_{00}^*(\Gamma) = \left(q_{00}\Sigma_{00}^{-1} + q_{10}\Sigma_{10}^{-1}\right)^{-1}\left(\Sigma_{00}^{-1}\mu_{00} + \Sigma_{10}^{-1}\mu_{10} - \Sigma_{10}^{-1}\Gamma(\mu_{11} - \mu_{01})\right) \quad \text{and}$$

$$\mu_{10}^*(\Gamma) = \left(q_{00}\Sigma_{00}^{-1} + q_{10}\Sigma_{10}^{-1}\right)^{-1}\left(\Sigma_{00}^{-1}\mu_{00} + \Sigma_{10}^{-1}\mu_{10} - \Sigma_{10}^{-1}\Gamma(\mu_{11} - \mu_{01})\right) + \Gamma(\mu_{11} - \mu_{01})$$

$$\Sigma_{00}^*(\Gamma) = (\Gamma\Sigma_{01}^{1/2})^2$$

$$\Sigma_{10}^*(\Gamma) = (\Gamma\Sigma_{11}^{1/2})^2.$$

By substituting these, when we look at the objective as a function of a positive semidefinite matrix-valued variable $\Gamma$, it turns out to be convex. This requires rewriting the expressions using the identities $\text{tr}(AB) = \text{tr}(BA), \det(AB) = \det(A)\det(B)$, and most importantly, $\text{tr}(AXBX) = \text{tr}\left((A^{1/2}XB^{1/2})(A^{1/2}XB^{1/2})^T\right)$ and $\log\det(AXBX) = \log\det\left(A^{1/2}XB^{1/2})(A^{1/2}XB^{1/2})^T\right)$, for symmetric, positive semidefinite matrices $A, B, X$ [58]. The convexity of the objective in $\Gamma$ follows from the convexity of $\text{tr}(AXBX)$ and $-\log\det(X)$ for matrix-valued variable $X$. Finally, we can solve it efficiently to get the optimal $\Gamma^*$. $\qquad\square$

**Corollary B.3.** *(Corollary 4.2 in the main text) For the case where $X|Y = i, A = a \sim \mathcal{N}(\mu_{ia}, \sigma_{ia}^2)$ are univariate normal distributions, for $i, a \in \{0, 1\}$, the optimal distribution $\tilde{D}$ from Theorem 4.1, with $\gamma^*$ being a function of the original distribution parameters, can be written down as:*

$$\tilde{\sigma}_{i0} = \gamma^*\sigma_{i1}, \quad \tilde{\mu}_{00} = \tilde{\mu}_{10} + \gamma^*(\mu_{01} - \mu_{11}), \text{ and } \tilde{\mu}_{10} = \frac{\left(q_{00}\dfrac{\mu_{00} - \gamma^*(\mu_{01} - \mu_{11})}{\sigma_{00}^2} + q_{10}\dfrac{\mu_{10}}{\sigma_{10}^2}\right)}{\left(\dfrac{q_{00}}{\sigma_{00}^2} + \dfrac{q_{10}}{\sigma_{10}^2}\right)},$$

*Proof.*

$$D_{\text{KL}}\left(\tilde{D}||D\right)$$

$$= \sum_{(x,i,a)} \Pr\left(\tilde{X}=x,\tilde{Y}=i,\tilde{A}=a\right) \log \frac{\Pr\left(\tilde{X}=x,\tilde{Y}=i,\tilde{A}=a\right)}{\Pr\left(X=x,Y=i,A=a\right)}$$

$$= \sum_{(x,i,a)} \Pr\left(\tilde{Y}=i,\tilde{A}=a\right)\Pr\left(\tilde{X}=x \mid \tilde{Y}=i,\tilde{A}=a\right)$$

$$\log \frac{\Pr\left(\tilde{Y}=i,\tilde{A}=a\right)\Pr\left(\tilde{X}=x \mid \tilde{Y}=i,\tilde{A}=a\right)}{\Pr\left(Y=y,A=a\right)\Pr\left(X=x \mid Y=i,A=a\right)}$$

$$= \sum_{(x,i,a)} \Pr\left(Y=i,A=a\right)\Pr\left(\tilde{X}=x \mid Y=i,A=a\right)$$

$$\log \frac{\Pr\left(Y=i,A=a\right)\Pr\left(\tilde{X}=x \mid Y=i,A=a\right)}{\Pr\left(Y=i,A=a\right)\Pr\left(X=x \mid Y=i,A=a\right)}$$

$$= \sum_{(i,a)} q_{ia} \sum_{x} \Pr\left(\tilde{X}=x \mid Y=i,A=a\right) \log \frac{\Pr\left(\tilde{X}=x \mid Y=i,A=a\right)}{\Pr\left(X=x \mid Y=i,A=a\right)}$$

$$= \sum_{(i,a)} q_{ia} D_{\text{KL}}\left(\tilde{P}_{ia}||P_{ia}\right)$$

$P_{ia}$ denotes the distribution of $X \mid Y=i, A=a \sim \mathcal{N}(\mu_{ia}, \sigma_{ia}^2)$ and $\tilde{P}_{ia}$ denotes the distribution of $\tilde{X} \mid Y=i, A=a \sim \mathcal{N}(\tilde{\mu}_{ia}, \tilde{\sigma}_{ia}^2)$. Their probability densities are

$$p_{ia}(x) = \frac{1}{x\sigma_{ia}\sqrt{2\pi}} \exp\left(-\frac{(x-\mu_{ia})^2}{2\sigma_{ia}^2}\right) \quad \text{and} \quad \tilde{p}_{ia}(x) = \frac{1}{x\tilde{\sigma}_{ia}\sqrt{2\pi}} \exp\left(-\frac{(x-\tilde{\mu}_{ia})^2}{2\tilde{\sigma}_{ia}^2}\right),$$

respectively. Hence,

$$D_{\text{KL}}\left(\tilde{P}_{ia}||P_{ia}\right) = \mathrm{E}\left[\log \frac{\tilde{p}_{ia}(\tilde{X})}{p_{ia}(\tilde{X})} \mid Y=i, A=a\right]$$

$$= \mathrm{E}\left[\frac{(\tilde{X}-\mu_{ia})^2}{2\sigma_{ia}^2} - \frac{(\tilde{X}-\tilde{\mu}_{ia})^2}{2\tilde{\sigma}_{ia}^2} + \log \frac{\sigma_{ia}}{\tilde{\sigma}_{ia}} \mid Y=i, A=a\right]$$

$$= \mathrm{E}\left[\left(\frac{1}{2\sigma_{ia}^2} - \frac{1}{2\tilde{\sigma}_{ia}^2}\right)\tilde{X}^2 + \left(\frac{\tilde{\mu}_{ia}}{\tilde{\sigma}_{ia}^2} - \frac{\mu_{ia}}{\sigma_{ia}^2}\right)\tilde{X} + \left(\frac{\mu_{ia}^2}{2\sigma_{ia}^2} - \frac{\tilde{\mu}_{ia}^2}{2\tilde{\sigma}_{ia}^2}\right) + \log \frac{\sigma_{ia}}{\tilde{\sigma}_{ia}} \mid Y=i, A=a\right]$$

$$= \left(\frac{1}{2\sigma_{ia}^2} - \frac{1}{2\tilde{\sigma}_{ia}^2}\right)(\tilde{\mu}_{ia}^2 + \tilde{\sigma}_{ia}^2) + \left(\frac{\tilde{\mu}_{ia}}{\tilde{\sigma}_{ia}^2} - \frac{\mu_{ia}}{\sigma_{ia}^2}\right)\tilde{\mu}_{ia} + \left(\frac{\mu_{ia}^2}{2\sigma_{ia}^2} - \frac{\tilde{\mu}_{ia}^2}{2\tilde{\sigma}_{ia}^2}\right) + \log \frac{\sigma_{ia}}{\tilde{\sigma}_{ia}}$$

$$= \frac{(\tilde{\mu}_{ia} - \mu_{ia})^2}{2\sigma_{ia}^2} + \frac{\tilde{\sigma}_{ia}^2 - \sigma_{ia}^2}{2\sigma_{ia}^2} + \log \frac{\sigma_{ia}}{\tilde{\sigma}_{ia}},$$

using $\mathrm{E}\left[\log \tilde{X} \mid Y=i, A=a\right] = \tilde{\mu}_{ia}$ and $\mathrm{E}\left[(\log \tilde{X})^2 \mid Y=i, A=a\right] = \tilde{\mu}_{ia}^2 + \tilde{\sigma}_{ia}^2$. Since we only change group $A=0$, we want to minimize

$$D_{\text{KL}}\left(\tilde{D}||D\right) = \sum_{i=0}^{1} q_{i0} D_{\text{KL}}\left(\tilde{P}_{i0}||P_{i0}\right)$$

$$= \sum_{i=0}^{1} q_{i0}\left(\frac{(\tilde{\mu}_{i0} - \mu_{i0})^2}{2\sigma_{i0}^2} + \frac{\tilde{\sigma}_{i0}^2 - \sigma_{i0}^2}{2\sigma_{i0}^2} + \log \frac{\sigma_{i0}}{\tilde{\sigma}_{i0}}\right)$$

as a function of the variables $\tilde{\mu}_{i0}$ and $\tilde{\sigma}_{i0}$ subject to the constraints

$$\frac{\mu_{01} - \mu_{11}}{\sigma_{11}} = \frac{\tilde{\mu}_{00} - \tilde{\mu}_{10}}{\tilde{\sigma}_{10}} \quad \text{and} \quad \frac{\sigma_{11}}{\sigma_{01}} = \frac{\tilde{\sigma}_{10}}{\tilde{\sigma}_{00}} \quad \text{and} \quad \tilde{\sigma}_{ia} \geq 0, \text{ for all } (i, a).$$

Let's fix $\gamma \in \mathbb{R}_{\geq 0}$ and minimize

$$\mathcal{L}_\gamma = \sum_{i=0}^{1} q_{i0} \left( \frac{(\tilde{\mu}_{i0} - \mu_{i0})^2}{2\sigma_{i0}^2} + \frac{\tilde{\sigma}_{i0}^2 - \sigma_{i0}^2}{2\sigma_{i0}^2} + \log \frac{\sigma_{i0}}{\tilde{\sigma}_{i0}} \right)$$

as a function of the variables $\tilde{\mu}_{ia}$ and $\tilde{\sigma}_{ia}$ subject to the following constraints

$$\frac{\tilde{\mu}_{00} - \tilde{\mu}_{10}}{\mu_{01} - \mu_{11}} = \frac{\tilde{\sigma}_{10}}{\sigma_{11}} = \frac{\tilde{\sigma}_{00}}{\sigma_{01}} = \gamma \quad \text{and} \quad \tilde{\sigma}_{ia} \geq 0, \text{ for all } (i, a).$$

The objective $\mathcal{L}_\gamma$ is convex and for a fixed $\gamma \in \mathbb{R}_{\geq 0}$, the constraints on are linear in $\tilde{\mu}_{i0}$ and $\tilde{\sigma}_{i0}$. Let's denote the optimal solution for a fixed $\gamma \in \mathbb{R}_{\geq 0}$ by $\mu_{i0}^*(\gamma)$ and $\sigma_{i0}^*(\gamma)$, for $i \in \{0, 1\}$. For a fixed $\gamma \in \mathbb{R}_{\geq 0}$, the above constraints fix $\sigma_{i0}^*(\gamma) = \gamma \sigma_{i1}$, for $i \in \{0, 1\}$, and by plugging in $\tilde{\mu}_{00} = \tilde{\mu}_{10} + \gamma(\mu_{01} - \mu_{11})$, we only need to minimize the following convex, quadratic objective in a single variable $\tilde{\mu}_{10}$,

$$\text{minimize} \quad q_{00} \frac{(\tilde{\mu}_{10} + \gamma(\mu_{01} - \mu_{11}) - \mu_{00})^2}{2\sigma_{00}^2} + q_{10} \frac{(\tilde{\mu}_{10} - \mu_{10})^2}{2\sigma_{10}^2}.$$

By equating the derivative to zero, we get the optimal solution as

$$\mu_{10}^*(\gamma) = \left( \frac{q_{00}}{\sigma_{00}^2} + \frac{q_{10}}{\sigma_{10}^2} \right)^{-1} \left( q_{00} \frac{\mu_{00} - \gamma(\mu_{01} - \mu_{11})}{\sigma_{00}^2} + q_{10} \frac{\mu_{10}}{\sigma_{10}^2} \right),$$

and the optimal value at $\mu_{10}^*(\gamma)$ is (The min of $ax^2 + bx + c$ occurs at $x = \frac{-b}{2a}$ and has value $c - \frac{b^2}{4a}$)

$$q_{00} \frac{(\gamma(\mu_{01} - \mu_{11}) - \mu_{00})^2}{2\sigma_{00}^2} + q_{10} \frac{\mu_{10}^2}{2\sigma_{10}^2} - \frac{1}{2} \frac{\left( q_{00} \frac{\mu_{00} - \gamma(\mu_{01} - \mu_{11})}{\sigma_{00}^2} + q_{10} \frac{\mu_{10}}{\sigma_{10}^2} \right)^2}{\left( \frac{q_{00}}{\sigma_{00}^2} + \frac{q_{10}}{\sigma_{10}^2} \right)}$$

$$= \frac{1}{2} \left( \frac{q_{00}}{\sigma_{00}^2} + \frac{q_{10}}{\sigma_{10}^2} \right)^{-1} \frac{q_{00} q_{10}}{\sigma_{00}^2 \sigma_{10}^2} \left( (\mu_{00} - \mu_{10}) - \gamma(\mu_{01} - \mu_{11}) \right)^2$$

$$= \frac{1}{2} \left( \frac{\sigma_{00}^2}{q_{00}} + \frac{\sigma_{10}^2}{q_{10}} \right)^{-1} \left( (\mu_{00} - \mu_{10}) - \gamma(\mu_{01} - \mu_{11}) \right)^2.$$

By plugging in the optimal solution, the minimum value of $\mathcal{L}_\gamma$ for a fixed $\gamma \in \mathbb{R}_{\geq 0}$ is given by

$$\mathcal{L}_\gamma^* = \sum_{i=0}^{1} q_{i0} \left( \frac{(\mu_{i0}^*(\gamma) - \mu_{i0})^2}{2\sigma_{i0}^2} + \frac{\sigma_{i0}^*(\gamma)^2 - \sigma_{i0}^2}{2\sigma_{i0}^2} + \log \frac{\sigma_{i0}}{\sigma_{i0}^*(\gamma)} \right)$$

$$= \frac{1}{2} \left( \frac{\sigma_{00}^2}{q_{00}} + \frac{\sigma_{10}^2}{q_{10}} \right)^{-1} \left( (\mu_{00} - \mu_{10}) - \gamma(\mu_{01} - \mu_{11}) \right)^2$$

$$+ q_{00} \frac{\gamma^2 \sigma_{01}^2 - \sigma_{00}^2}{2\sigma_{00}^2} + q_{10} \frac{\gamma^2 \sigma_{11}^2 - \sigma_{10}^2}{2\sigma_{10}^2} + (q_{00} + q_{10}) \log \frac{1}{\gamma} + q_{00} \log \frac{\sigma_{00}}{\sigma_{01}} + q_{10} \log \frac{\sigma_{10}}{\sigma_{11}}$$

$$= \frac{1}{2} \left( \frac{\sigma_{00}^2}{q_{00}} + \frac{\sigma_{10}^2}{q_{10}} \right)^{-1} \left( (\mu_{00} - \mu_{10}) - \gamma(\mu_{01} - \mu_{11}) \right)^2 + \frac{q_{00}}{2} \left( \gamma^2 \frac{\sigma_{01}^2}{\sigma_{00}^2} - 1 \right)$$

$$+ (q_{00} + q_{10}) \log \frac{1}{\gamma} + q_{00} \log \frac{\sigma_{00}}{\sigma_{01}} + q_{10} \log \frac{\sigma_{10}}{\sigma_{11}}.$$

This is a convex objective in $\gamma$ (because the second derivative is non-negative) and by equating the derivative to zero, we have that the optimal $\gamma^*$ must satisfy

$$\frac{(\mu_{01} - \mu_{11}) \left( \gamma^*(\mu_{01} - \mu_{11}) - (\mu_{00} - \mu_{10}) \right)}{\left( \frac{\sigma_{00}^2}{q_{00}} + \frac{\sigma_{10}^2}{q_{10}} \right)} + \gamma^* \left( q_{00} \frac{\sigma_{01}^2}{\sigma_{00}^2} + q_{10} \frac{\sigma_{11}^2}{\sigma_{10}^2} \right) - \frac{q_{00} + q_{10}}{\gamma^*} = 0.$$

Multiplying with $\gamma^* \left( \frac{\sigma_{00}^2}{q_{00}} + \frac{\sigma_{10}^2}{q_{10}} \right)$, we can write it as a quadratic equation as follows.

$$\left( (\mu_{01} - \mu_{11})^2 + \sigma_{01}^2 + \sigma_{11}^2 + \frac{q_{10}\sigma_{00}^2}{q_{00}\sigma_{10}^2}\sigma_{11}^2 + \frac{q_{00}\sigma_{10}^2}{q_{10}\sigma_{00}^2}\sigma_{01}^2 \right) \gamma^{*2}$$

$$-(\mu_{01} - \mu_{11})(\mu_{00} - \mu_{10})\gamma^* - (q_{00} + q_{10}) \left( \frac{\sigma_{00}^2}{q_{00}} + \frac{\sigma_{10}^2}{q_{10}} \right) = 0$$

The discriminant of the above quadratic polynomial is non-negative because the leading coefficient is positive and the constant term is negative. So this polynomial has two real roots. Moreover, since the constant term is negative, it cannot have both positive or both negative roots. Its only non-negative root is the optimal solution $\gamma^* \in \mathbb{R}_{\geq 0}$ we want.

$$\gamma^* = \frac{(\mu_{01} - \mu_{11})(\mu_{00} - \mu_{10}) + \sqrt{\Delta}}{2 \left( (\mu_{01} - \mu_{11})^2 + \sigma_{01}^2 + \sigma_{11}^2 + \frac{q_{10}\sigma_{00}^2}{q_{00}\sigma_{10}^2}\sigma_{11}^2 + \frac{q_{00}\sigma_{10}^2}{q_{10}\sigma_{00}^2}\sigma_{01}^2 \right)}, \text{ where}$$

$$\Delta = (\mu_{01} - \mu_{11})^2(\mu_{00} - \mu_{10})^2$$

$$+ 4 \left( (\mu_{01} - \mu_{11})^2 + \sigma_{01}^2 + \sigma_{11}^2 + \frac{q_{10}\sigma_{00}^2}{q_{00}\sigma_{10}^2}\sigma_{11}^2 + \frac{q_{00}\sigma_{10}^2}{q_{10}\sigma_{00}^2}\sigma_{01}^2 \right) (q_{00} + q_{10}) \left( \frac{\sigma_{00}^2}{q_{00}} + \frac{\sigma_{10}^2}{q_{10}} \right).$$

$\square$

**Proposition B.4.** *(Proof of Proposition 4.3 in the main text) Let $(X, Y, A)$ denote the features, binary class label, and binary group membership, respectively, of a random data point from any data distribution $D$ with $q_{ia} = \Pr(Y = i, A = a)$, for $i \in \{0, 1\}$ and $a \in \{0, 1\}$, such that $q_{10}/q_{00} = q_{11}/q_{01}$, and let $X|Y = i, A = a \sim \mathcal{N}(\mu_{ia}, \sigma_{ia}^2)$ be univariate normal distributions, for $i \in \{0, 1\}$ and $a \in \{0, 1\}$. Let $\tilde{D}$ denote a distribution obtained by keeping $(Y, A)$ unchanged and only changing $X|Y = i, A = a$ to $\tilde{X}|Y = i, A = a \sim \mathcal{N}(\tilde{\mu}_{ia}, \tilde{\sigma}_{ia}^2)$. Then minimizing $D_{\mathrm{KL}}\left(\tilde{D}||D\right)$ as a function of the variables $\tilde{\mu}_{ia}$ and $\tilde{\sigma}_{ia}$ subject to the constraints in Proposition 3.2 leads to a non-convex program. Furthermore, let $\gamma^* = \arg\min_{\gamma \in (0, \infty)} \mathcal{L}_\gamma^*$ for some non-convex function of $\gamma$ that is only dependent on the original distribution parameters. Then, all the new distribution parameters $\tilde{\mu}_{ia}$ and $\tilde{\sigma}_{ia}$ can be expressed as a function of $\gamma^*$ and the original distribution parameters $\mu_{ia}$ and $\sigma_{ia}$.*

*Proof.* We consider the following optimization program

$$D_{\mathrm{KL}}\left(\tilde{D}||D\right) = \sum_{(i,a)} q_{ia} D_{\mathrm{KL}}\left(\tilde{P}_{ia}||P_{ia}\right)$$

$$= \sum_{(i,a)} q_{ia} \left( \frac{(\tilde{\mu}_{ia} - \mu_{ia})^2}{2\sigma_{ia}^2} + \frac{\tilde{\sigma}_{ia}^2 - \sigma_{ia}^2}{2\sigma_{ia}^2} + \log \frac{\sigma_{ia}}{\tilde{\sigma}_{ia}} \right)$$

as a function of the variables $\tilde{\mu}_{ia}$ and $\tilde{\sigma}_{ia}$ subject to the constraints

$$\frac{\tilde{\mu}_{01} - \tilde{\mu}_{11}}{\tilde{\sigma}_{11}} = \frac{\tilde{\mu}_{00} - \tilde{\mu}_{10}}{\tilde{\sigma}_{10}} \quad \text{and} \quad \frac{\tilde{\sigma}_{11}}{\tilde{\sigma}_{01}} = \frac{\tilde{\sigma}_{10}}{\tilde{\sigma}_{00}} \quad \text{and} \quad \tilde{\sigma}_{ia} \geq 0, \text{ for all } (i, a).$$

Let's fix $\gamma \in \mathbb{R}_{\geq 0}$ and minimize

$$\mathcal{L}_\gamma = \sum_{(i,a)} q_{ia} \left( \frac{(\tilde{\mu}_{ia} - \mu_{ia})^2}{2\sigma_{ia}^2} + \frac{\tilde{\sigma}_{ia}^2 - \sigma_{ia}^2}{2\sigma_{ia}^2} + \log \frac{\sigma_{ia}}{\tilde{\sigma}_{ia}} \right)$$

as a function of the variables $\tilde{\mu}_{ia}$ and $\tilde{\sigma}_{ia}$ subject to the following constraints

$$\frac{\tilde{\mu}_{01} - \tilde{\mu}_{11}}{\tilde{\mu}_{00} - \tilde{\mu}_{10}} = \frac{\tilde{\sigma}_{11}}{\tilde{\sigma}_{10}} = \frac{\tilde{\sigma}_{01}}{\tilde{\sigma}_{00}} = \gamma \quad \text{and} \quad \tilde{\sigma}_{ia} \geq 0, \text{ for all } (i, a).$$

Now the objective $\mathcal{L}_\gamma$ is convex and for a fixed $\gamma \in \mathbb{R}_{\geq 0}$, the constraints on are linear in $\tilde{\mu}_{ia}$ and $\tilde{\sigma}_{ia}$. Let's denote the optimal solution for a fixed $\gamma \in \mathbb{R}_{\geq 0}$ by $\mu_{ia}^*(\gamma)$ and $\sigma_{ia}^*(\gamma)$, for $i, a \in \{0, 1\}$. To

find this, we can split the above objective into parts that can be optimized separately as follows.

$$\text{minimize} \quad \sum_{(i,a)} q_{ia} \frac{(\tilde{\mu}_{ia} - \mu_{ia})^2}{2\sigma_{ia}^2} \quad \text{subject to} \quad \tilde{\mu}_{01} - \tilde{\mu}_{11} = \gamma(\tilde{\mu}_{00} - \tilde{\mu}_{10}), \quad \text{and}$$

$$\text{minimize} \quad \sum_{(i,a)} q_{ia} \left( \frac{\tilde{\sigma}_{ia}^2 - \sigma_{ia}^2}{2\sigma_{ia}^2} + \log \frac{\sigma_{ia}}{\tilde{\sigma}_{ia}} \right) \quad \text{subject to} \quad \tilde{\sigma}_{i1} = \gamma\tilde{\sigma}_{i0}, \text{and } \tilde{\sigma}_{ia} \geq 0, \text{ for all } (i,a).$$

For each $i \in \{0,1\}$, by substituting $\tilde{\sigma}_{i1} = \gamma\tilde{\sigma}_{i0}$, we need to optimize a function in only one variable $\tilde{\sigma}_{i0}$. The optimal solutions $\sigma_{ia}^*(\gamma)$ turn out to be

$$\sigma_{i0}^*(\gamma) = \sqrt{\frac{q_{i0} + q_{i1}}{\dfrac{q_{i0}}{\sigma_{i0}^2} + \dfrac{q_{i1}\gamma^2}{\sigma_{i1}^2}}} \quad \text{and} \quad \sigma_{i1}^*(\gamma) = \gamma\sqrt{\frac{q_{i0} + q_{i1}}{\dfrac{q_{i0}}{\sigma_{i0}^2} + \dfrac{q_{i1}\gamma^2}{\sigma_{i1}^2}}}, \quad \text{for } i \in \{0,1\},$$

Now let's find the optimal solutions $\mu_{ia}^*(\gamma)$. The gradient of the objective must be parallel to the linear constraint, so

$$\frac{q_{00}(\mu_{00}^*(\gamma) - \mu_{00})}{\sigma_{00}^2} = -\gamma\lambda, \; \frac{q_{01}(\mu_{01}^*(\gamma) - \mu_{01})}{\sigma_{01}^2} = \lambda,$$

$$\frac{q_{10}(\mu_{10}^*(\gamma) - \mu_{10})}{\sigma_{10}^2} = \gamma\lambda, \; \frac{q_{11}(\mu_{11}^*(\gamma) - \mu_{11})}{\sigma_{11}^2} = -\lambda,$$

for some $\lambda \in \mathbb{R}$, which gives

$$\mu_{00}^*(\gamma) = -\gamma\lambda\frac{\sigma_{00}^2}{q_{00}} + \mu_{00}, \quad \mu_{01}^*(\gamma) = \lambda\frac{\sigma_{01}^2}{q_{01}} + \mu_{01},$$

$$\mu_{10}^*(\gamma) = \gamma\lambda\frac{\sigma_{10}^2}{q_{10}} + \mu_{10}, \quad \mu_{11}^*(\gamma) = -\lambda\frac{\sigma_{11}^2}{q_{11}} + \mu_{11}.$$

Since $\mu_{ia}^*(\gamma)$ satisfies the constraint $\dfrac{\tilde{\mu}_{01} - \tilde{\mu}_{11}}{\tilde{\mu}_{00} - \tilde{\mu}_{10}} = \gamma$, we have

$$\frac{\lambda\dfrac{\sigma_{01}^2}{q_{01}} + \mu_{01} + \lambda\dfrac{\sigma_{11}^2}{q_{11}} - \mu_{11}}{-\gamma\lambda\dfrac{\sigma_{00}^2}{q_{00}} + \mu_{00} - \gamma\lambda\dfrac{\sigma_{10}^2}{q_{10}} - \mu_{10}} = \gamma, \quad \text{and hence,} \quad \lambda = \frac{\gamma(\mu_{00} - \mu_{10}) - (\mu_{01} - \mu_{11})}{\dfrac{\sigma_{01}^2}{q_{01}} + \dfrac{\sigma_{11}^2}{q_{11}} + \gamma^2\left(\dfrac{\sigma_{00}^2}{q_{00}} + \dfrac{\sigma_{10}^2}{q_{10}}\right)}.$$

Thus, we can express $\mu_{ia}^*(\gamma)$ as

$$\mu_{00}^*(\gamma) = -\gamma\frac{\gamma(\mu_{00} - \mu_{10}) - (\mu_{01} - \mu_{11})}{\dfrac{\sigma_{01}^2}{q_{01}} + \dfrac{\sigma_{11}^2}{q_{11}} + \gamma^2\left(\dfrac{\sigma_{00}^2}{q_{00}} + \dfrac{\sigma_{10}^2}{q_{10}}\right)} \frac{\sigma_{00}^2}{q_{00}} + \mu_{00}$$

$$\mu_{01}^*(\gamma) = \frac{\gamma(\mu_{00} - \mu_{10}) - (\mu_{01} - \mu_{11})}{\dfrac{\sigma_{01}^2}{q_{01}} + \dfrac{\sigma_{11}^2}{q_{11}} + \gamma^2\left(\dfrac{\sigma_{00}^2}{q_{00}} + \dfrac{\sigma_{10}^2}{q_{10}}\right)} \frac{\sigma_{01}^2}{q_{01}} + \mu_{01}$$

$$\mu_{10}^*(\gamma) = \gamma\frac{\gamma(\mu_{00} - \mu_{10}) - (\mu_{01} - \mu_{11})}{\dfrac{\sigma_{01}^2}{q_{01}} + \dfrac{\sigma_{11}^2}{q_{11}} + \gamma^2\left(\dfrac{\sigma_{00}^2}{q_{00}} + \dfrac{\sigma_{10}^2}{q_{10}}\right)} \frac{\sigma_{10}^2}{q_{10}} + \mu_{10}$$

$$\mu_{11}^*(\gamma) = -\frac{\gamma(\mu_{00} - \mu_{10}) - (\mu_{01} - \mu_{11})}{\dfrac{\sigma_{01}^2}{q_{01}} + \dfrac{\sigma_{11}^2}{q_{11}} + \gamma^2\left(\dfrac{\sigma_{00}^2}{q_{00}} + \dfrac{\sigma_{10}^2}{q_{10}}\right)} \frac{\sigma_{11}^2}{q_{11}} + \mu_{11}.$$

Thus, the optimal value of $\mathcal{L}_\gamma$ for a fixed $\gamma \in \mathbb{R}_{\geq 0}$ is given by

$$\mathcal{L}_\gamma^* = \sum_{(i,a)} q_{ia} \left( \frac{(\mu_{ia}^*(\gamma) - \mu_{ia})^2}{2\sigma_{ia}^2} + \frac{\sigma_{ia}^*(\gamma)^2 - \sigma_{ia}^2}{2\sigma_{ia}^2} + \log \frac{\sigma_{ia}}{\sigma_{ia}^*(\gamma)} \right).$$

Dividing the above expression into three parts, the first part evaluates to

$$
\sum_{(i,a)} q_{ia} \frac{(\mu_{ia}^*(\gamma) - \mu_{ia})^2}{2\sigma_{ia}^2} = \frac{q_{00}}{2\sigma_{00}^2} \frac{\gamma^2 \lambda^2 \sigma_{00}^4}{q_{00}^2} + \frac{q_{01}}{2\sigma_{01}^2} \frac{\lambda^2 \sigma_{01}^4}{q_{01}^2} + \frac{q_{10}}{2\sigma_{10}^2} \frac{\gamma^2 \lambda^2 \sigma_{10}^4}{q_{10}^2} + \frac{q_{11}}{2\sigma_{11}^2} \frac{\lambda^2 \sigma_{11}^4}{q_{11}^2}
$$

$$
= \frac{\gamma^2 \lambda^2 \sigma_{00}^2}{2q_{00}} + \frac{\lambda^2 \sigma_{01}^2}{2q_{01}} + \frac{\gamma^2 \lambda^2 \sigma_{10}^2}{2q_{10}} + \frac{\lambda^2 \sigma_{11}^2}{2q_{11}}
$$

$$
= \frac{\lambda^2}{2} \left( \frac{\sigma_{01}^2}{q_{01}} + \frac{\sigma_{11}^2}{q_{11}} + \gamma^2 \left( \frac{\sigma_{00}^2}{q_{00}} + \frac{\sigma_{10}^2}{q_{10}} \right) \right)
$$

$$
= \frac{1}{2} \left( \frac{\gamma(\mu_{00} - \mu_{10}) - (\mu_{01} - \mu_{11})}{\frac{\sigma_{01}^2}{q_{01}} + \frac{\sigma_{11}^2}{q_{11}} + \gamma^2 \left( \frac{\sigma_{00}^2}{q_{00}} + \frac{\sigma_{10}^2}{q_{10}} \right)} \right)^2 \left( \frac{\sigma_{01}^2}{q_{01}} + \frac{\sigma_{11}^2}{q_{11}} + \gamma^2 \left( \frac{\sigma_{00}^2}{q_{00}} + \frac{\sigma_{10}^2}{q_{10}} \right) \right)
$$

$$
= \frac{1}{2} \frac{(\gamma(\mu_{00} - \mu_{10}) - (\mu_{01} - \mu_{11}))^2}{\frac{\sigma_{01}^2}{q_{01}} + \frac{\sigma_{11}^2}{q_{11}} + \gamma^2 \left( \frac{\sigma_{00}^2}{q_{00}} + \frac{\sigma_{10}^2}{q_{10}} \right)}.
$$

The second part evaluates to

$$
\sum_{(i,a)} q_{ia} \frac{\sigma_{ia}^*(\gamma)^2 - \sigma_{ia}^2}{2\sigma_{ia}^2} = \sum_{(i,a)} \frac{q_{ia}}{2} \left( \frac{\sigma_{ia}^*(\gamma)^2}{\sigma_{ia}^2} - 1 \right)
$$

$$
= \sum_{i=0}^{1} \frac{q_{i0}}{2} \left( \frac{q_{i0} + q_{i1}}{\sigma_{i0}^2 \left( \frac{q_{i0}}{\sigma_{i0}^2} + \frac{q_{i1}\gamma^2}{\sigma_{i1}^2} \right)} - 1 \right) + \sum_{i=0}^{1} \frac{q_{i1}}{2} \left( \frac{\gamma^2(q_{i0} + q_{i1})}{\sigma_{i1}^2 \left( \frac{q_{i0}}{\sigma_{i0}^2} + \frac{q_{i1}\gamma^2}{\sigma_{i1}^2} \right)} - 1 \right)
$$

$$
= \sum_{i=0}^{1} \frac{q_{i0}}{2} \frac{q_{i1} \left( 1 - \frac{\sigma_{i0}^2 \gamma^2}{\sigma_{i1}^2} \right)}{q_{i0} + q_{i1} \frac{\sigma_{i0}^2 \gamma^2}{\sigma_{i1}^2}} + \sum_{i=0}^{1} \frac{q_{i1}}{2} \frac{q_{i0} \left( \gamma^2 - \frac{\sigma_{i1}^2}{\sigma_{i0}^2} \right)}{q_{i0} \frac{\sigma_{i1}^2}{\sigma_{i0}^2} + q_{i1}\gamma^2}
$$

$$
= \sum_{i=0}^{1} \frac{q_{i0}}{2} \frac{q_{i1} \left( 1 - \frac{\sigma_{i0}^2 \gamma^2}{\sigma_{i1}^2} \right)}{q_{i0} + q_{i1} \frac{\sigma_{i0}^2 \gamma^2}{\sigma_{i1}^2}} + \sum_{i=0}^{1} \frac{q_{i1}}{2} \frac{q_{i0} \left( \frac{\sigma_{i0}^2 \gamma^2}{\sigma_{i1}^2} - 1 \right)}{q_{i0} + q_{i1} \frac{\sigma_{i0}^2 \gamma^2}{\sigma_{i1}^2}}
$$

$$
= 0,
$$

and the third part evaluates to

$$
\sum_{(i,a)} q_{ia} \log \frac{\sigma_{ia}}{\sigma_{ia}^*(\gamma)} = \sum_{i=0}^{1} \frac{q_{i0}}{2} \log \frac{\sigma_{i0}^2}{\sigma_{i0}^*(\gamma)^2} + \frac{q_{i1}}{2} \log \frac{\sigma_{i1}^2}{\sigma_{i1}^*(\gamma)^2}
$$

$$
= \sum_{i=0}^{1} \frac{q_{i0}}{2} \log \frac{\sigma_{i0}^2 \left( \frac{q_{i0}}{\sigma_{i0}^2} + \frac{q_{i1}\gamma^2}{\sigma_{i1}^2} \right)}{q_{i0} + q_{i1}} + \frac{q_{i1}}{2} \log \frac{\sigma_{i1}^2 \left( \frac{q_{i0}}{\sigma_{i0}^2} + \frac{q_{i1}\gamma^2}{\sigma_{i1}^2} \right)}{\gamma^2(q_{i0} + q_{i1})}
$$

$$
= \sum_{i=0}^{1} \frac{q_{i0}}{2} \log \frac{\frac{q_{i0}}{q_{i1}} + \gamma^2 \frac{\sigma_{i0}^2}{\sigma_{i1}^2}}{\frac{q_{i0}}{q_{i1}} + 1} + \frac{q_{i1}}{2} \log \frac{\frac{q_{i0}}{q_{i1}} + \gamma^2 \frac{\sigma_{i0}^2}{\sigma_{i1}^2}}{\gamma^2 \frac{\sigma_{i0}^2}{\sigma_{i1}^2} \left( \frac{q_{i0}}{q_{i1}} + 1 \right)}
$$

$$
= \sum_{i=0}^{1} \frac{q_{i0} + q_{i1}}{2} \log \left( \frac{q_{i0}}{q_{i1}} + \gamma^2 \frac{\sigma_{i0}^2}{\sigma_{i1}^2} \right) - \frac{q_{i0} + q_{i1}}{2} \log \left( \frac{q_{i0}}{q_{i1}} + 1 \right) - q_{i1} \log \gamma - q_{i1} \log \frac{\sigma_{i0}}{\sigma_{i1}}.
$$

Putting it all together

$$\mathcal{L}_\gamma^* = \sum_{(i,a)} q_{ia} \left( \frac{(\mu_{ia}^*(\gamma) - \mu_{ia})^2}{2\sigma_{ia}^2} + \frac{\sigma_{ia}^*(\gamma)^2 - \sigma_{ia}^2}{2\sigma_{ia}^2} + \log \frac{\sigma_{ia}}{\sigma_{ia}^*(\gamma)} \right)$$

$$= \frac{1}{2} \frac{(\gamma(\mu_{00} - \mu_{10}) - (\mu_{01} - \mu_{11}))^2}{\frac{\sigma_{01}^2}{q_{01}} + \frac{\sigma_{11}^2}{q_{11}} + \gamma^2 \left( \frac{\sigma_{00}^2}{q_{00}} + \frac{\sigma_{10}^2}{q_{10}} \right)} + \sum_{i=0}^1 \frac{q_{i0} + q_{i1}}{2} \log \left( \frac{q_{i0}}{q_{i1}} + \gamma^2 \frac{\sigma_{i0}^2}{\sigma_{i1}^2} \right)$$

$$- \frac{q_{i0} + q_{i1}}{2} \log \left( \frac{q_{i0}}{q_{i1}} + 1 \right) - q_{i1} \log \gamma - q_{i1} \log \frac{\sigma_{i0}}{\sigma_{i1}}.$$

Minimizing $\mathcal{L}_\gamma^*$ leads to a non convex program. Since $\gamma$ is the ratio between variances of the new subgroup distribution, for a practical solution, we can do a line search over $\gamma \in (0, B)$ for some $B < \infty$. $\qquad \square$

**Bound on Unfairness and Error Rate** For completeness, we now derive upper bounds on the error rate and the unfairness gap $\Delta_{\text{EO}}$ of the Bayes optimal classifier $\tilde{h}$ on $\tilde{D}$ with respect to the original distribution $D$. These bounds show that both the accuracy loss and the fairness gap depend only on the KL divergence between $D$ and $\tilde{D}$. It also shows that the optimal value of our optimization problem can be used to approximately translate the accuracy guarantee of $\tilde{h}$ from $\tilde{D}$ to $D$.

**Proposition B.5.** *(Proposition 4.4 in the main text) Let $err(h, D)$ denote the error rate (expected 0-1 loss) of a classifier $h$ on the distribution $D$. Let $d_{TV}(\tilde{D}, D)$ denote the total variation distance between two distributions $\tilde{D}$ and $D$, while $D_{KL}$ denotes the KL-Divergence between them. Denote the Bayes optimal classifier on the ideal distribution $\tilde{D}$ as $\tilde{h}$ (and similarly the Bayes optimal classifier $h$. Then, we can bound the error rate and Equal opportunity of $\tilde{h}$ on the original distribution $D$ as follows:*

$$|err(\tilde{h}, D) - err(\tilde{h}, \tilde{D})| \leq \sqrt{2 D_{KL}(\tilde{D}, D)} \quad \text{and} \quad \Delta_{EO}(\tilde{h}, D) \leq \sqrt{8 D_{\text{KL}} \left( \tilde{D} \| D \right)}.$$

*Proof.* For the sake of this proof, we assume a countable data domain. Using the definition of the expected $0 - 1$ loss, we can write:

$$\text{err}(\tilde{h}, D) - \text{err}(\tilde{h}, \tilde{D})$$
$$= \sum_{(x,y,a)} \mathbb{I}\left( \tilde{h}(x,a) \neq y \right) \cdot (p(x,y,a) - \tilde{p}(x,y,a))$$
$$\leq 2 d_{TV}(\tilde{D}, D) \leq \sqrt{2 D_{KL}(\tilde{D}, D)} \qquad \text{(Pinsker's Inequality [19])}.$$

The first inequality follows from writing the error as expected 0-1 loss and using the definition of total variation distance. The last line follows from Pinsker's inequality [19]. We can similarly prove the other direction to obtain the first inequality. Similarly, for the true positive rate of group $a$, $TPR_a(\tilde{h}, D) - TPR_a(\tilde{h}, \tilde{D}) \leq 2 D_{TV}(\tilde{D}, D)$.

We can also write for the other group $A = a'$, $TPR_{a'}(\tilde{h}, \tilde{D}) - TPR_{a'}(\tilde{h}, D) \leq 2 D_{TV}(\tilde{D}, D)$. Adding both LHS and RHS and repeating the exercise in the other direction, noting that the TPR difference of $\tilde{h}$ in $\tilde{D}$ is zero (since in the ideal distribution we have exact fairness), we can bound the absolute value of the TPR difference, which is our definition of $\Delta_{EO}$, we get:

$$\Delta_{EO}(\tilde{h}, D) \leq \sqrt{8 D_{\text{KL}} \left( \tilde{D} \| D \right)}$$

$\qquad \square$

**Equalizing the first moment** A popular intervention in the fairness literature is to equalize the first moment of the two sensitive groups or the mean outcomes of two groups, also known as the

Calders-Verwer gap [17, 44, 22]. We, therefore, also study an intervention where we only change the mean of the under-privileged group and try to match it with the mean of the privileged group. We can show that the resulting optimization program is convex. We leverage this intervention in Section 5 of the main text.

**Proposition B.6.** *(Affirmative Action by Equalizing First Moments) Let $(X, Y, A)$ denote the features, binary class label, and binary group membership, respectively, of a random data point from any data distribution $D$ with $q_{ia} = \Pr(Y = i, A = a)$, for $i \in \{0, 1\}$ and $a \in \{0, 1\}$. Let $X|Y = i, A = a \sim \mathcal{N}(\mu_{ia}, \sigma_{ia}^2)$ be a univariate Normal distribution, for $i \in \{0, 1\}$ and $a \in \{0, 1\}$. Then in the case of Affirmative mean change, where we impose the following constraints:*

$$\frac{q_{10}\,\tilde{\mu}_{10}}{q_{10} + q_{00}} + \frac{q_{00}\,\tilde{\mu}_{00}}{q_{10} + q_{00}} = \frac{q_{11}\,\mu_{11}}{q_{11} + q_{01}} + \frac{q_{01}\,\mu_{01}}{q_{11} + q_{01}},$$

*we can efficiently minimize $D_{\mathrm{KL}}\left(\tilde{D}||D\right)$ as a function of the variables $\tilde{\mu}_{i0}$ and $\tilde{\Sigma}_{i0}$.*

*Proof.* We are dealing with the following optimization problem:

$$D_{\mathrm{KL}}\left(\tilde{D}||D\right) = \sum_{i=0}^{1} q_{i0} D_{\mathrm{KL}}\left(\tilde{P}_{i0}||P_{i0}\right)$$

$$= \sum_{i=0}^{1} q_{i0}\left(\frac{(\tilde{\mu}_{i0} - \mu_{i0})^2}{2\sigma_{i0}^2} + \frac{\tilde{\sigma}_{i0}^2 - \sigma_{i0}^2}{2\sigma_{i0}^2} + \log\frac{\sigma_{i0}}{\tilde{\sigma}_{i0}}\right)$$

as a function of the variables $\tilde{\mu}_{i0}$ and $\tilde{\sigma}_{i0}$ subject to the constraints

$$\frac{q_{10}}{q_{10} + q_{00}}\tilde{\mu}_{10} + \frac{q_{00}}{q_{10} + q_{00}}\tilde{\mu}_{00} = \frac{q_{11}}{q_{11} + q_{01}}\mu_{11} + \frac{q_{01}}{q_{11} + q_{01}}\mu_{01}$$

Since we are only changing the means and keeping the variances the same, the objective only depends on $\tilde{\mu}_{i0}$. Furthermore, let $K = {(q_{10}+q_{00})}/{(q_{11}+q_{01})} \cdot (q_{11}\mu_{11} + q_{01}\mu_{01})$ so that

$$\mathcal{L} = \sum_{i=0}^{1} q_{i0}\frac{(\tilde{\mu}_{i0} - \mu_{i0})^2}{2\sigma_{i0}^2}, \quad \text{subject to } \tilde{\mu}_{00} = \frac{K - \tilde{\mu}_{10}}{q_{00}}.$$

Substituting the constraint on $\tilde{\mu}_{00}$ in the objective $\mathcal{L}$ gives us a convex quadratic in $\tilde{\mu}_{10}$, and the solution is obtained by setting the derivative to zero:

$$\tilde{\mu}_{00} = \frac{\frac{K}{\sigma_{10}^2 \cdot q_{10}} - \frac{\mu_{10}}{\sigma_{10}^2} + \frac{\mu_{00}}{\sigma_{00}^2}}{\frac{q_{00}}{\sigma_{10}^2 \cdot q_{10}} + \frac{1}{\sigma_{00}^2}}, \quad \tilde{\mu}_{10} = \frac{\frac{K}{\sigma_{00}^2 \cdot q_{00}} - \frac{\mu_{00}}{\sigma_{00}^2} + \frac{\mu_{10}}{\sigma_{10}^2}}{\frac{q_{10}}{\sigma_{00}^2 \cdot q_{00}} + \frac{1}{\sigma_{10}^2}}, \quad \tilde{\mu}_{01} = \mu_{01}, \quad \text{and} \quad \tilde{\mu}_{11} = \mu_{11}.$$

$\square$

## C    Additional Figures for Section 5

In this section, we lay out additional plots from our Gaussian case study. We first describe the setup. We modify a stylized setting of Gaussian distributions from previous work (see Definition 3.1 in [59], Section 5.3 in [4]) to investigate the unfairness and the Bayes optimal error on the original and ideal distributions obtained through various interventions. We fix $q_{ia} \in (0, 1)$ such that $q_{00} + q_{10} + q_{01} + q_{11} = 1$, and our data generation works as follows. We simulate a data distribution where $Y = i, A = a$ with probability $q_{ia}$ and $X \mid Y = i, A = a$ is sampled from a univariate Gaussian $\mathcal{N}(\mu_{ia}, \sigma_{ia}^2)$. We choose homoskedastic Gaussians within each group $A = a$, i.e., $\sigma_{0a} = \sigma_{1a}$, so the we can show the Bayes optimal classifier boundary as a threshold. We choose different $\sigma_{ia}$'s that cover ground truth distribution that can the entire spectrum of being *ideal* or close to *ideal* to very far, and then we apply different interventions to change all or some subset of $\mu_{ia}$'s and $\sigma_{ia}$'s to find the nearest *ideal* distribution in KL-divergence as given in Section 4 of the main text.

We first look at a case where the subgroup distributions are the same shifted versions of each other in Figure 5. Note that all interventions, in this case, result in the same Bayes error (BE), but affirmative

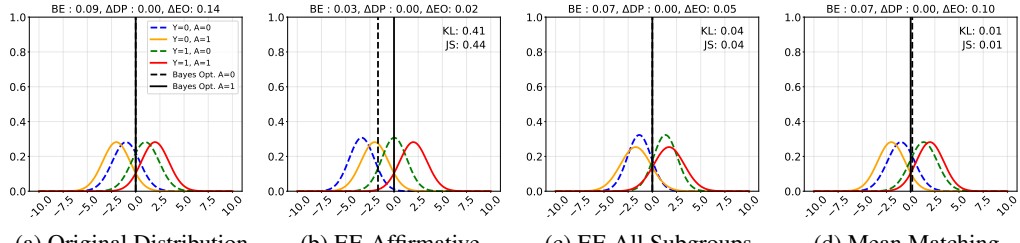

(a) Original Distribution    (b) EF-Affirmative    (c) EF-All Subgroups    (d) Mean Matching

Figure 5: Comparison of Different Interventions when the subgroup distributions are shifted version of each other. While all methods achieve the same Bayes Error, Affirmative action is able to bring down the Bayes Error and achieve exact fairness.

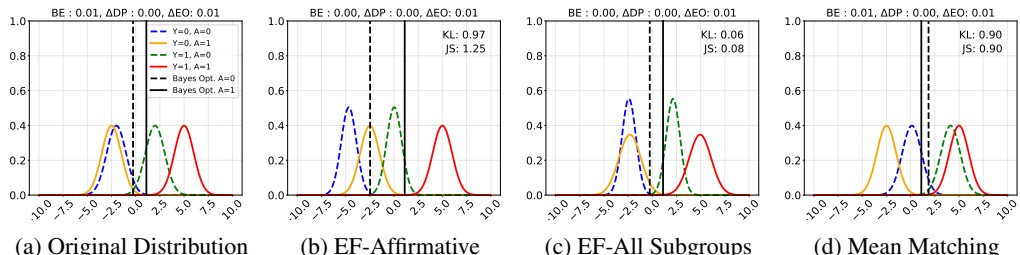

(a) Original Distribution    (b) EF-Affirmative    (c) EF-All Subgroups    (d) Mean Matching

Figure 6: Comparison of Different Interventions when the original distribution is already fair. In this case, EF-All ensures that it stays close to the true distribution, as no intervention as required, while others relatively deviate.

action brings the BE down with zero unfairness at the cost of incurring a deviation in terms of KL and JS divergence. However, in the next subplot, changing all four subgroups not only helps reduce the Bayes error and unfairness but also stays very close to the true distribution in the KL/JS sense. Matching the means also helps reduce the unfairness while staying close to the true distribution, but is sub-optimal compared to the EF-Affirmative and EF-All interventions.

Next, we look at a case where the Bayes optimal classifier is already fair ($\Delta$EO is close to $0$ while $\Delta$DP=$0$) in Figure 6. The expected solution here should be that any intervention must leave the distribution as it is. EF-Affirmative intervention keeps the unfairness and error rate numbers as it is, but deviates from the true distribution, as indicated by the KL/JS divergences. However, the EF-All intervention only makes major changes to variances and stays close to the true distribution. The Mean Matching intervention shifts both the under-privileged subgroups and strays away from the true distribution, as indicated by relatively high KL/JS values.

Finally, in light of Proposition 3.3, we simulate the cost-sensitive risk for a different cost matrix $C$ other than 0-1 loss by considering a threshold $t_C = 3/4$ on $\eta(x, a)$ in Figure 7. The original distribution has high unfairness. EF-Affirmative intervention manages to achieve almost perfect fairness and zero error rate, but incurs relatively high KL/JS numbers. However, once again, changing

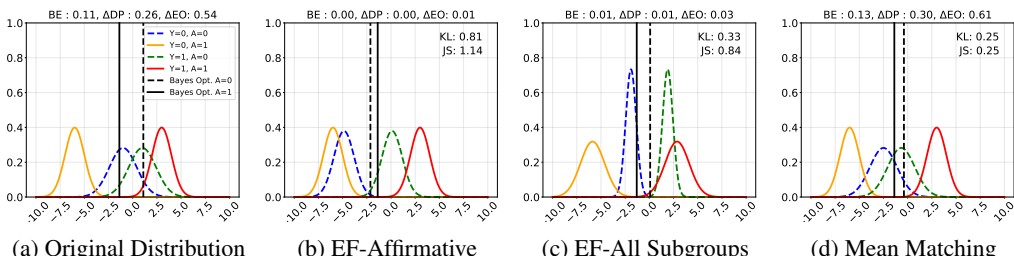

(a) Original Distribution    (b) EF-Affirmative    (c) EF-All Subgroups    (d) Mean Matching

Figure 7: Comparison of Different Interventions when we use a different threshold ($3/4$) than the Bayes optimal threshold ($1/2$). As derived in Proposition 3.3, the EF-Affirmative and EF-All interventions work with any threshold.

all four subgroups, results in a solution that is perfectly fair and accurate, with low KL/JS. Mean Matching is unable to address the fairness-accuracy tension at all in this case and also manages to drift away from the true distribution, as indicated by non-zero KL/JS values.

# D  Details and Additional Results for Section 6

In this section, we lay down all the details for the experiments performed for LLM steering. The code to reproduce our results is provided *here*. For the multi-class experiments, we use a lot of helper functions from the code of Singh et al. [67] [2]. For the emotion steering experiments, we reproduce the methodology from Zhao et al. [79] and provide the Jupyter notebook in our code.

## D.1  Reducing Disparity in Multi-class classification

To apply our intervention for multi-class settings, we first come up with a version of Theorem 4.1 for multiple classes. We show this for a univariate distribution, and for our intervention, we assume diagonal covariance. Since our experiment setup only requires two groups for each class, we show the effective constraints assuming two sensitive groups, but this methodology can be readily extended to handle a countable number of groups as well. To make our program convex (and affirmative), we fix a class $y \in \mathcal{Y}$. We fix our class $y^*$ according to the following heuristic: $y^* = \underset{y \in \mathcal{Y}}{\arg\min} \, \Delta_y TPR(\hat{h})$, where $\hat{h}$ is the empirical risk minimizer on the given data. This fixes our ratio $\gamma_\sigma = \frac{\sigma_{y^*1}}{\sigma_{y^*0}}$ and $\gamma_q = \frac{q_{y^*1}}{q_{y^*0}}$.

We can now write a multi-class version of the optimization program in Theorem 4.1:

$$\mathcal{L}_\gamma = \sum_{(i,a)} q_{ia} \left( \frac{(\tilde{\mu}_{ia} - \mu_{ia})^2}{2\sigma_{ia}^2} + \frac{\tilde{\sigma}_{ia}^2 - \sigma_{ia}^2}{2\sigma_{ia}^2} + \log \frac{\sigma_{ia}}{\tilde{\sigma}_{ia}} \right)$$

as a function of the variables $\tilde{\mu}_{ia}$ and $\tilde{\sigma}_{ia}$ subject to the following constraints

$$\frac{\tilde{\mu}_{i1} - \tilde{\mu}_{j1}}{\tilde{\mu}_{i0} - \tilde{\mu}_{j0}} = \frac{\tilde{\sigma}_{i1}}{\tilde{\sigma}_{i0}} = \frac{\tilde{\sigma}_{j1}}{\tilde{\sigma}_{j0}} = \gamma_\sigma \,, \frac{q_{i1}}{q_{i0}} = \frac{q_{j1}}{q_{j0}} = \gamma_q \quad \text{and} \quad \tilde{\sigma}_{ia} \geq 0, \text{ for all } i \in \mathcal{Y}, j \in \mathcal{Y} \setminus \{i\}.$$

Just like in the proof of Theorem 4.1, the resulting program will result in separable objectives for a class $y$ and then in the underlying optimization variables $\tilde{\mu}_{ya}$ and $\tilde{\sigma}_{ya}$:

**Program for $\tilde{\mu}_{ya}$:**

$$q_{y0} \frac{(\tilde{\mu}_{y0} - \mu_{y0})^2}{2\sigma_{y0}^2} + q_{y1} \frac{(\tilde{\mu}_{y1} - \mu_{y1})^2}{2\sigma_{y1}^2}, \text{ subject to } \tilde{\mu}_{y1} - \tilde{\mu}_{y^*1} = \gamma(\tilde{\mu}_{y0} - \tilde{\mu}_{y^*0})$$

**Program for $\tilde{\sigma}_{ya}$:**

$$q_{y0} \left( \frac{\tilde{\sigma}_{y0}^2 - \sigma_{y0}^2}{2\sigma_{y0}^2} + \log \frac{\sigma_{y0}}{\tilde{\sigma}_{y0}} \right) + q_{y1} \left( \frac{\tilde{\sigma}_{y1}^2 - \sigma_{y1}^2}{2\sigma_{y1}^2} + \log \frac{\sigma_{y1}}{\tilde{\sigma}_{y1}} \right), \text{ subject to } \tilde{\sigma}_{y1} = \gamma\tilde{\sigma}_{y0},$$

where $y^*$ is the class we fixed earlier and $\gamma = \gamma_\sigma$. The solution for the following programs are the following:

$$\tilde{\sigma}_{y0} = \pm \sqrt{\frac{q_{y0} + q_{y1}}{\frac{q_{y0}}{\sigma_{y0}^2} + \frac{\gamma^2 q_{y1}}{\sigma_{y1}^2}}}, \ \tilde{\sigma}_{y1} = \pm\gamma \sqrt{\frac{q_{y0} + q_{y1}}{\frac{q_{y0}}{\sigma_{y0}^2} + \frac{\gamma^2 q_{y1}}{\sigma_{y1}^2}}}, \tilde{\mu}_{y0} = \frac{\frac{q_{y0}}{\sigma_{y0}^2}\mu_{y0} + \frac{\gamma q_{y1}}{\sigma_{y1}^2}(\mu_{y1} - \mu_{y^*1} + \gamma\mu_{y^*0})}{\frac{q_{y0}}{\sigma_{y0}^2} + \frac{\gamma^2 q_{y1}}{\sigma_{y1}^2}}, \text{ and } \tilde{\mu}_{y1} =$$

$$\frac{\frac{q_{y0}}{\sigma_{y0}^2}(\mu_{y^*1} - \gamma\mu_{y^*0} + \gamma\mu_{y0}) + \frac{\gamma^2 q_{y1}}{\sigma_{y1}^2}\mu_{y1}}{\frac{q_{y0}}{\sigma_{y0}^2} + \frac{\gamma^2 q_{y1}}{\sigma_{y1}^2}}.$$

Once we have the corrected distributions $\mathcal{N}(\tilde{\mu}_{ia}, \tilde{\Sigma}_{ia})$, we set up an affine intervention, following the design choices of Singh et al. [67]. We assume an affine relationship between the original

---

[2] https://github.com/shauli-ravfogel/affine-steering

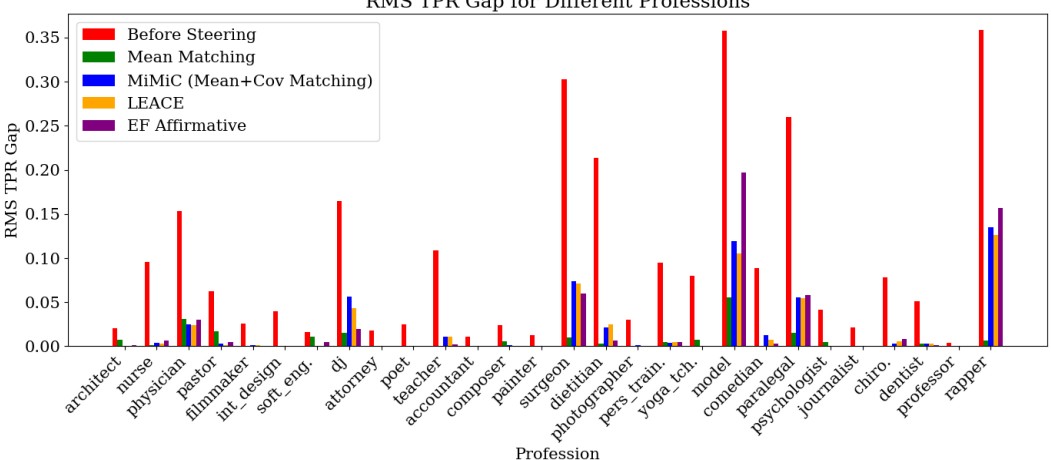

Figure 8: TPR-gap between Gender groups for all professions. All methods to steer feature representations achieve roughly the same accuracy (in the range of 0.77-0.79). Our intervention (EF Affirmative) is able to significantly reduce the TPR-gap for all professions. In many cases, it is even comparable or better than previous interventions Belrose et al. [8], Singh et al. [67].

and transformed samples per subgroup: $Y = a_{ya}X + b_{ya}$, where $Y \sim \mathcal{N}(\tilde{\mu}_{ya}, \tilde{\sigma}_{ya})$ and $X \sim \mathcal{N}(\mu_{ya}, \sigma_{ya})$. Taking expectation on both sides gives us: $\tilde{\mu}_{ya} = a_{ya}\mu_{ya} + b_{ya}, \tilde{\sigma}_{ya}^2 = a_{ya}^2\sigma_{ya}^2$, and we get the following coefficients: $a_{ya} = \pm\frac{\tilde{\sigma}_{ya}}{\sigma_{ya}}$ and $b_{ya} = \tilde{\mu}_{ya} \pm \frac{\tilde{\sigma}_{ya}}{\sigma_{ya}}\mu_{ya}$.

We have two choices of the parameters corresponding to the positive and negative solutions. Since we are working with empirical estimates, we use the validation error to decide the best set of estimates. A detailed implementation is given in the code. To implement the conditions for $q_{ya}$, we use the reweighing scheme of Kamiran and Calders [43]. The plot in the main text (Figure 3) assumes $q_{y0} = q_{y1}$ for the corrected covariances. Figure 8 shows the plot with no such assumption for $q_{ya}$ and confirms with same trends as observed in Figure 3.

## D.2 Steering activations for Joyful generation

Zhao et al. [79] propose to obtain a distribution over the steering vectors for a concept instead of a single steering vector. In this section, we lay out all our prompts and the design choices for the emotion steering pipeline.

We first generate training data for each concept (joyful, angry) for each of the groups (horror, comedy), resulting in four subgroups. The following prompt was used to generate an initial set of data:

> **Prompt to generate 1000 Comedy movie reviews that are joyful.**
>
> Compose a concise 30-word movie review, assuming it is a comedy movie, that covers these four aspects: plot, sound and music, cultural impact, and emotional resonance. Choose a joyful tone for your review. For the plot, comment on its structure or originality. Regarding sound and music, mention how it enhances the storytelling. For cultural impact, touch on any relevant social commentary. Finally, describe how the film resonates emotionally. Ensure your joyful tone is consistent throughout the review. Please include emotions like "joyful" in these texts and generate 1000 samples.

We obtain the last token embeddings from each layer of a Llama-3.1 8B model [36] for each of the samples. We now proceed towards obtaining the steering vectors. We want to obtain a steering vector for each group. Treating angry reviews as an irrelevant sample for the 'joyful' concept, we assign $y = 0$ to angry samples and $y = 1$ to joyful samples. Because we want to estimate a distribution over the steering vectors instead of a single vector, we sample 300 points with replacement and repeat this for 50 iterations. In each of these iterations, we train a Logistic Regression model to classify between

the relevant and the irrelevant samples. We get 50 weight vectors using this pipeline, and we use those to obtain the sample mean and covariance. We denote the resulting distribution for the steering vectors for layer $l$ as $\mathcal{N}(\mu_{1a}^l, \Sigma_{1a}^l)$ where $1a$ denotes that this distribution represents the steering vector for joyful emotion for a group $A = a$ (horror or comedy reviews). To apply our intervention later, we also obtain the steering vector in the other direction by flipping the relevance labels, i.e. joyful $\rightarrow$ angry, and we denote the resulting Gaussian distribution with $\mathcal{N}(\mu_{0a}^l, \Sigma_{0a}^l)$.

To perform steering, we now sample a steering vector $v_c^l \sim \mathcal{N}(\mu_{1a}^l, \Sigma_{1a}^l)$ and add it to the last token representation of layer $l$ with strength $a$: $h^l = (1 - a)h^l + av_c^l$. To measure the performance of steering, we ask the Llama model to generate angry reviews using the following prompt:

> **Prompt to evaluate the joyfulness of generated reviews (for group horror).**
>
> **System Prompt:** You are a well informed movie critic.
> **User Prompt:** Compose a concise 10-word movie review, assuming it is a horror movie. The tone of the review must be angry. The review should be within 10 words. Generate 100 samples. Please make sure that you do not repeat the same review and keep each review under 10 words. Also, make sure that the reviews are coherent and not repetitive.

We first measure the joyfulness score of the generated reviews without adding any steering vectors. Just like Zhao et al. [79], we use the following prompt to ask the GPT 4.1 model to rate the joyfulness of the text:

**System Prompt:** You are a deterministic evaluation agent trained to assess text samples using two strictly defined criteria: (1) emotional tone (joyful score) and (2) linguistic structure (chaotic score).

For each input, your task is to produce exactly two integer scores between 0 and 10. These must follow the user's scoring rubric precisely. The first score reflects emotional positivity or dark exuberance without considering sentence structure. The second score reflects linguistic disorder or repetition, without considering sentiment.

You must produce output in CSV format, strictly matching the following structure: "sample", joyful score, chaotic score

Do not include any additional content, commentary, or variation. Your outputs must be fully deterministic: identical input always produces identical output.

Your responses should match the tone and evaluation logic demonstrated in the provided reference examples.

**User Prompt:** You are an automated evaluator tasked with rating horror movie reviews using two fixed criteria: **joyful expression** and **textual coherence and novelty**.

For each review, you must generate exactly two integer scores as described below:

—

### **1. Joyful Score**
Rate from 0 to 10 based strictly on emotional tone:
* **0** = not joyful at all (strongly negative, angry, or emotionally flat tone)
* **10** = extremely joyful, enthusiastic, or darkly exuberant in tone
* Intermediate values reflect a mix of enthusiasm and criticism
* Focus solely on affect and sentiment; ignore linguistic structure for this score

—

### **2. Chaotic Score**
Rate from 0 to 10 based strictly on linguistic organisation:
* **0** = fully coherent, fluent, and structurally sound
* **10** = highly chaotic, incoherent, repetitive, or nonsensical
* Intermediate values reflect partial disorganisation, verbosity, or broken phrasing
* Focus only on syntax, flow, and novelty of expression; ignore emotional content

—

### **Reference Examples**
**Joyful and Coherent Horror Reviews:**
* "Blood-soaked fun ensues in this delightfully terrifying slasher film."
* "Chilling thrills abound in this creepy haunted mansion tale."
* "Jump scares galore in this electrifying horror comedy gem."
* "Unsettling unease fills this unnerving psychological horror masterpiece."
* "Bone-chilling chills chill to the bone in this one."

**Angry and Coherent Horror Reviews:**
* "Abysmal plot twists ruined what could've been a decent film."
* "Mind-numbing terror fails to deliver in this lazy horror."
* "Weak jump scares can't save this trainwreck disaster."
* "Poor production values ruin what little suspense exists."
* "Frustratingly predictable, making it boring and unscary too."

—

### **Output Format**
* For each sample, return one line in strict CSV format: '"sample", joyful score, chaotic score'
* **Example Output:**
'' sample, joyful score, chaotic score
"This horror film was painfully dull and predictable.", joyful_score_1, chaotic_score_1
"Terrifying, stylish, and packed with chilling moments!", joyful_score_2, chaotic_score_2
'''

* Do **not** include explanations, commentary, or additional formatting.
* Output must be **fully deterministic**: the same input must always yield the same scores.
Begin processing the dataset now. Here is the batch of review samples:

A few notes on evaluation are in order. Zhao et al. [79] report both joyfulness and coherence scores for the generated text. However, we observed that coherence scores were all over the place and did not make sense. Second, Zhao et al. evaluate using the GPT-4o model, whereas we used the GPT 4.1 model since we observed that the joyful scores corroborated more with the qualitative inspection of the generated samples.

Following the above pipeline, we observe an increase in joyfulness scores of the generated reviews by a Llama model after steering. However, since the effectiveness of joyful steering was not the same for the horror and comedy movie review generations, we apply our affirmative intervention (Theorem 4.1), assuming that the horror and comedy movie reviews define two groups. Let the modified steering vector be denoted by $\tilde{v}_c^l \sim \mathcal{N}(\tilde{\mu}_{1a}^l, \tilde{\Sigma}_{1a}^l)$, where the new gaussian distribution is obtained after applying the affirmative action intervention from Theorem 4.1 assuming horror group is the under-privileged group.

However, simply replacing $v_c^l$ will not work. We demonstrate that empirically in the main text, where in Figure 4, $\alpha = 1$ corresponds to using $\tilde{v}_c^l$ instead of $v_c^l$. But we can always use $\tilde{v}_c^l$ to nudge the existing steering vector $v_c^l$ in the right direction. To do that, we modify the steering vector and the representation $h^l$ with the following rule: $h^l = (1 - a)h^l + a((1 - \alpha)v_c^l + \alpha\tilde{v}_c^l)$, where $\alpha$ controls the strength of mixing the old and new steering vectors. In Figure 4, we show that for small values of $\alpha$, the steering vector indeed starts performing better in steering the reviews of the horror group towards a more joyful tone.

