# OpenReview forum: "On Optimal Steering to Achieve Exact Fairness"
_NeurIPS.cc/2025/Conference — NeurIPS 2025 poster_

### Official Review · Reviewer_YLZ4 · 2025-06-07

**Clarity:** 3
**Significance:** 2
**Originality:** 3
**Rating:** 5
**Confidence:** 3

**Summary:**

This paper proposed a preprocessing method to achieve fair classification by steering the feature distribution into an ideal distribution where all cost-sensitive Bayes optimal classifiers will ensure equal opportunity (EO) (also other metrics under a univariate Gaussian setting). The paper first defined the ideal distribution and characterized its property when the distribution is multivariate/univariate Gaussian. Then it formulated the feature steering as an optimization problem to minimize the KL divergence between the target distribution and the ideal distribution $\tilde{D}$. Since the KL divergence constraint is convex and finding $\tilde{D}$ can be tractable for certain distribution families, we can solve for $D$ in practice. Numerical experiments demonstrated the effectness of the method.

**Questions:**

Could you provide the trade-off results on accuracy?

**Ethical Concerns:**

["NO or VERY MINOR ethics concerns only"]

**Final Justification:**

I encourage the authors to add the accuracy results and the discussions on accuracy trade-offs.

Currently, I will retain my score and see whether there are some other points raised in reviewer AC discussion phase.

**Limitations:**

yes

**Quality:**

3

**Strengths And Weaknesses:**

> Strengths

- The problem formulation is clear and well-motivated.
- The theoretical results are plausible.
- The LLM experiments are interestiing.
- The writing is clear.

> Weaknesses

- Most theoretical results of finding $\tilde{D}$ in the main paper are based on Gaussian distribution. And the conditions are only sufficient for multivariate Gaussian. Also, the discussed classifiers are Bayes optimal ones. I understand it is somehow standard in theoretical papers, but not sure of the practical applicability. Is it possible to discuss more general distributions?

- Theorem 4.1 and Prop 4.3 demonstrate how we can find $D$, but the discussions on accuracy guarantee are deferred to Appendix (prop 2.5). However, this trade-off can be very important while understanding whether the preprocessing methods should be used in practice. I suggest discussing this more in the main paper.

- Similarly, in the experiment results of section 6, it would be better to show the accuracy comparison (though it is mentioned that accuracy of EF  Affirmative is at the same level of other methods).


> Minor: I think the literature on long-term fairness can also be related to this work. For example, "steering" and long-term fairness in performative prediction happens naturally when agents respond to the decision-maker [1,2]. It would be interesting to consider whether your preprocessing method can also benefit long-term fairness.

[1] Hardt M, Jagadeesan M, Mendler-Dünner C. Performative power[J]. Advances in Neural Information Processing Systems, 2022, 35: 22969-22981.

[2] Jin, Kun, et al. "Addressing Polarization and Unfairness in Performative Prediction." arXiv preprint arXiv:2406.16756 (2024).

---

> ### Author Rebuttal · Authors · 2025-07-31
>
> Thank you for your careful reading and valuable feedback. We answer your questions below.
>
> 1. **Could you provide the trade-off results on accuracy?**: On page 12 of the supplementary material, Proposition 2.5 shows how to bound the change in accuracy in terms of the KL divergence between the original distribution and the ideal distribution. Interestingly, this change can be a drop in accuracy in the worst case, but could also be a gain in the best case, unlike the accuracy traded off for fairness when we do not change the underlying data distribution.
>
> We also comment on the some points raised in the Weakness section below:
>
> 1. **Most theoretical results of finding in the main paper are based on Gaussian distribution. And the conditions are only sufficient for multivariate Gaussian. Also, the discussed classifiers are Bayes optimal ones. I understand it is somehow standard in theoretical papers, but not sure of the practical applicability. Is it possible to discuss more general distributions?**: You are correct in pointing that out. The fairness of the Bayes optimal classifier, and the corresponding optimization programs to search for the ideal distributions, are easier to mathematically analyze for parametric families of distributions, e.g., multivariate Gaussians. Multivariate Gaussians may not always fit real-world data but can be justifiably used to model concepts and internal representations in machine learning, as shown in prior work [3,4]. For future work, we can move towards general distributions by considering a class of distributions with bounded moments [5] or working with invertible transformations to parametric families [6].
>
> 2. **Theorem 4.1 and Prop 4.3 demonstrate how we can find $D$, but the discussions on accuracy guarantee are deferred to Appendix (prop 2.5). However, this trade-off can be very important while understanding whether the preprocessing methods should be used in practice. I suggest discussing this more in the main paper.**: We will definitely include it in the main manuscript in further revisions of the paper and discuss the implications of this result.
>
> 3. **Similarly, in the experiment results of section 6, it would be better to show the accuracy comparison (though it is mentioned that accuracy of EF Affirmative is at the same level of other methods).** Please find the accuracy numbers for all the methods in Section 6.1, where we reduced the disparity in multi-class classification -> Before steering: 0.789, Mean Matching: 0.788, MiMiC (Mean+Cov Matching): 0.793, LEACE: 0.788, EF Affirmative: 0.779. We will also include them in the main manuscript in further revisions.
>
> 4. **Minor: I think the literature on long-term fairness can also be related to this work. For example, "steering" and long-term fairness in performative prediction happens naturally when agents respond to the decision-maker [1,2]. It would be interesting to consider whether your preprocessing method can also benefit long-term fairness.**: This is certainly a very interesting direction, and we will definitely comment on this as a potential future work in further revisions. Thank you so much for suggesting this direction.
>
> We greatly appreciate your positive feedback and request you to champion our paper for acceptance.
>
> [1] Hardt M, Jagadeesan M, Mendler-Dünner C. Performative power[J]. Advances in Neural Information Processing Systems, 2022, 35: 22969-22981.
>
> [2] Jin, Kun, et al. "Addressing Polarization and Unfairness in Performative Prediction." arXiv preprint arXiv:2406.16756 (2024).
>
> [3] Zhao, Haiyan, et al. "Beyond single concept vector: Modeling concept subspace in llms with gaussian distribution." arXiv preprint arXiv:2410.00153 (2024)
>
> [4] Eftekhari, Daniel, and Vardan Papyan. "On the Importance of Gaussianizing Representations." arXiv preprint arXiv:2505.00685 (2025).
>
> [5] Murti, Chaitanya, and Chiranjib Bhattacharyya. "DisCEdit: Model Editing by Identifying Discriminative Components." Advances in Neural Information Processing Systems 37 (2024): 47261-47296.
>
> [6] Theisen, Ryan, et al. "Evaluating state-of-the-art classification models against bayes optimality." Advances in Neural Information Processing Systems 34 (2021): 9367-9377.

---

> > ### Comment · Reviewer_YLZ4 · 2025-07-31
> > **Thanks for the rebuttal**
> >
> > I encourage the authors to add the accuracy results and the discussions on accuracy trade-offs.
> >
> > Currently, I will retain my score and see whether there are some other points raised in reviewer AC discussion phase.

---

> > > ### Author Response · Authors · 2025-08-03
> > >
> > > As per your suggestion, we will move Proposition 2.5 from the supplementary to the main paper and add relevant discussion. Thank you once again for your prompt response and positive feedback.

---

### Official Review · Reviewer_c89k · 2025-06-19

**Clarity:** 2
**Significance:** 2
**Originality:** 3
**Rating:** 3
**Confidence:** 3

**Summary:**

This paper first explores an ideal distribution under which group-aware classifiers can achieve absolute fairness metrics, along with the corresponding conditions required to reach such a distribution. The authors then propose a method that solves a KL divergence problem to approximate this ideal distribution. The results show that both the “Affirmative” approach and altering all subgroups lead to significant fairness improvements, while also enhancing classification performance. Based on this theory, the authors test LLMs on two tasks to validate their practical effectiveness in real-world scenarios.

**Questions:**

1. On the definition of the ideal distribution, I don't understand why DP can also be incorporated. As the optimal classifier and DP can be in conflict.

2. How is it possible to make the prediction both fair and accurate? Does it mean the previous Pareto frontier works [1, 2] are not meaningful?


[1] Kim, Joon Sik, Jiahao Chen, and Ameet Talwalkar. "Fact: A diagnostic for group fairness trade-offs." International Conference on Machine Learning. PMLR, 2020.

[2] Xian, Ruicheng, Lang Yin, and Han Zhao. "Fair and optimal classification via post-processing." International conference on machine learning. PMLR, 2023.

**Ethical Concerns:**

["NO or VERY MINOR ethics concerns only"]

**Final Justification:**

After the rebuttal, I believe some of my questions have been well addressed. I actually don't quite get the constraint of this theoretical work, i.e. there must be some constraint on the optimization procedure. But from my perspective, it has a relatively free assumption about data distribution in the discussion for fairness and performance. Therefore, I still lean towards a borderline score.

**Limitations:**

Yes

**Paper Formatting Concerns:**

I don't have such a concern.

**Quality:**

2

**Strengths And Weaknesses:**

Strengths:

**1.**  The author finds the condition for ideal distribution.

**2.** It is theoretically guaranteed to achieve EO fairness.

Weakness: I have read this paper at least three times, below are the weaknesses I found.

**1** Although this paper is theoretically based, its presentation lacks rigor. For example, in line 122, it introduces $h(\cdot)$ without a definition. Line 148 should be argmin $L$, not the probability. Line 127, the second Probability expression, etc., makes it very hard to check the correctness.

**2.** I also found this theory's condition is relatively strong, as the condition introduced in line 171, I don't believe real-world data can fit such a condition.

**3.** It also assumes univariate Gaussion on condition $X$ in Proposition 3.3, Theroem 4.1. It is an oversimplification of the data distribution.

**4.** The theoretical part is based on a Bayesian optimal classifier. I don’t quite understand the experiment, which involves an LLM-based representation steering task.

**5** Did the author assume that Test and Training are IID?

---

> ### Author Rebuttal · Authors · 2025-07-31
>
> Thank you for carefully reading our paper and providing your valuable feedback. We address your concerns below.
>
> 1. **On the definition of the ideal distribution, I don't understand why DP can also be incorporated. As the optimal classifier and DP can be in conflict.**: We clarify the definition of ideal distribution and correct the typos you pointed out in lines 127 and 148.
> Definition 2.1 for Demographic Parity (line 122) and lines 126-127 should read as: $$\Delta_{DP, y}(h, D) = \max_{a, a' \in \mathcal{A}} |Pr(h(X, A)=y|A=a) - Pr(h(X, A)=y|A=a')|.$$
> We can explicitly define $l_C$ in line 136 as: Let $C \in \mathbb{R}^{|\mathcal{Y}| \times |\mathcal{Y}|}$ be a matrix that defines the cost-sensitive loss function $l_{C}$ of predicting label $i$ when the true label is $y$ as $l_C(h(x,a), y) = c_{h(x, a) y}$, the $(h(x, a), y)$-th entry of $C$. The cost-sensitive risk of a classifier $h$ is then given by $E_{D}[l_{C} (h(X,A), Y)]$.
> Finally, Definition 3.1 should be: Let $D$ be a distribution over $\mathcal{X} \times \mathcal{A} \times \mathcal{Y}$. Let $\mathcal{H}$ be a hypothesis class of group-aware classifiers $h: \mathcal{X} \times \mathcal{A} \rightarrow \mathcal{Y}$, and let $h^* = \underset{h \in \mathcal{H}}{\arg\min}~ E_{D}[l_{C}(h(X,A), Y)]$, where $(X,A,Y) \sim D$. We call $D$ an ideal distribution if $\Delta_{DP, y} ( h^{\*} , D)=0$, for all $y \in \mathcal{Y}$ (similarly, $\Delta_{EO, y}(h^{\*}, D)=0$, for all $y \in \mathcal{Y}~$).
> Propositions 3.3 and 3.2 show conditions under which many univariate and multivariate Gaussians provably give ideal distributions for Equal Opportunity (i.e., equal TPRs across groups). The exact same idea works for equality of FPRs and gives equality of positive rates across groups, or equivalently, Demographic Parity (DP).
>
> 2. **How is it possible to make the prediction both fair and accurate?**: This observation is not new, as several prior works demonstrate that the fairness-accuracy trade-off disappears when data bias is accounted for [3,4,5,6,7,8]. Our Theorem 4.1 essentially gives an algorithmic recipe to minimally change a given data distribution and provably achieve this.
>
> 3. **Does it mean the previous Pareto frontier works [1, 2] are not meaningful?**: Pareto-frontier or fairness-accuracy trade-offs are meaningful when the underlying data distribution is fixed. [1,2] maximize accuracy subject to various fairness constraints to explore this trade-off, but do not change the data distribution.
>
> We will now address some of the points raised in the Weakness Section.
>
> 1. **Presentation lacks rigor. For example, line 122, 148, 127 make it very hard to check the correctness.**: Thank you for pointing out these typos. We will correct all of them and have explained our corrections above in our response to your question 1. Our proofs are correct, and I hope our explanation helps you verify the same.
>
> 2. **I don't believe real-world data can fit such a condition [as in line 171].**: We do not model real-world data using Gaussians. Our empirical work applies Gaussian modeling for concepts and internal representations where it is justified by previous work [9,10]. The condition in line 171 can be simplified further to f-divergence as follows. If $\Sigma_{ia} = \Sigma_a$ (same group covariance for all classes), then our condition is equivalent to having the same f-divergence between any two class-conditional distributions across different groups. In other words, $\Delta_{\Sigma_{a}}^{2}(\mu_{ia}, \mu_{ja})=  \Delta_{\Sigma_ {a'}}^{2}(\mu_{ia'}, \mu_{ja'})$, $\forall i,j \in \mathcal{Y}, a,a' \in \mathcal{A}$; see f-divergence formula (P1) near the end of page 2 in arxiv: 2204.10952. f-divergence is commonly used in fair classification and generation.
>
> 3. **It also assumes univariate Gaussian on condition in Proposition 3.3, Theorem 4.1. It is an oversimplification of the data distribution.**  Theorem 4.1 and Proposition 3.2 are stated for multivariate Gaussian distributions. Proposition 3.3 and Corollary 4.2 for univariate Gaussians are only to help the reader as special cases.
>
> 4. **The theoretical part is based on a Bayesian optimal classifier. I don’t quite understand the experiment, which involves an LLM-based representation steering task.**: Our theoretical results are on the fairness of the Bayes optimal classifier, which is easier to mathematically analyse for parametric families of distributions, e.g., multivariate Gaussians. Multivariate Gaussians may not always fit real-world data, but can be justifiably used to model concepts and internal representations in machine learning, as shown in prior work [9,10]. Hence, we felt that LLM-based representation-steering makes a compelling application for our technique.
>
> 5. **Did the author assume that Test and Training are IID?**: Yes. It is a standard assumption in Machine Learning.
>
> We sincerely thank you for reading our paper carefully. We request you to kindly revisit our paper with our detailed explanations above and upgrade your rating.
>
> [1] Kim, Joon Sik, Jiahao Chen, and Ameet Talwalkar. "Fact: A diagnostic for group fairness trade-offs." International Conference on Machine Learning. PMLR, 2020.
>
> [2] Xian, Ruicheng, Lang Yin, and Han Zhao. "Fair and optimal classification via post-processing." International conference on machine learning. PMLR, 2023.
>
> [3] Wick, Michael, and Jean-Baptiste Tristan. "Unlocking fairness: a trade-off revisited." Advances in neural information processing systems 32 (2019).
>
> [4] Blum, Avrim, and Kevin Stangl. "Recovering from biased data: Can fairness constraints improve accuracy?." arXiv preprint arXiv:1912.01094 (2019).
>
> [5] Dutta, Sanghamitra, et al. "Is there a trade-off between fairness and accuracy? a perspective using mismatched hypothesis testing." International conference on machine learning. PMLR, 2020.
>
> [6] Maity, Subha, et al. "Does enforcing fairness mitigate biases caused by subpopulation shift?." Advances in Neural Information Processing Systems 34 (2021): 25773-25784.
>
> [7] Sharma, Mohit, and Amit Deshpande. "How far can fairness constraints help recover from biased data?." arXiv preprint arXiv:2312.10396 (2023).
>
> [8] Leininger, Charlotte, Simon Rittel, and Ludwig Bothmann. "Overcoming Fairness Trade-offs via Pre-processing: A Causal Perspective." arXiv preprint arXiv:2501.14710 (2025).
>
> [9] Zhao, Haiyan, et al. "Beyond single concept vector: Modeling concept subspace in llms with gaussian distribution." arXiv preprint arXiv:2410.00153 (2024)
>
> [10] Eftekhari, Daniel, and Vardan Papyan. "On the Importance of Gaussianizing Representations." arXiv preprint arXiv:2505.00685 (2025).

---

> > ### Comment · Reviewer_c89k · 2025-08-02
> > **Thank you for your rebuttal**
> >
> > The reviewer sincerely thanks the authors for the detailed response. I am still confused about some questions. I would like to share with the authors and other reviewers if they can provide some insights for me.
> >
> > (1) Still related to Demographic Parity: This metric highly depends on the base rate of a set, i.e. $pr(y=1|a)$ [1]. In other words, DP may not be 0 even if you have a 100% accuracy classifier. There can be no $h*$ to both minimize the cost function and achieve perfect DP fairness.
> >
> > (2) The question I ask for the assumption for training and test IID is because some fairness issues are rooted in subpopulation shifts, for example, spurious correlation. Such a shift can result in the fair representation learning failing.
> >
> > (3) Even after reading through the rebuttal, I still feel that Proposition 3.3 is a very strong assumption.
> >
> >
> > [1] Zhao, Han. "Fair and optimal prediction via post‐processing." AI Magazine 45.3 (2024): 411-418.

---

> > > ### Author Response · Authors · 2025-08-03
> > > **Responses to Follow-up Questions**
> > >
> > > Thank you for carefully reading our rebuttal. Here are our responses to your follow-up questions.
> > >
> > > 1. **Still related to Demographic Parity: This metric highly depends on the base rate of a set, i.e. $pr(y=1|a)$ [1]. In other words, DP may not be 0 even if you have a 100% accuracy classifier. There can be no $h^*$ to both minimize the cost function and achieve perfect DP fairness.**:  When $y = h^* (x, a)$ or the Bayes optimal classifier $h^* $ has $100\%$ accuracy, please see the following condition in Proposition 3.3:
> > > $$\frac{\text{Pr}(Y=1, A=0)}{\text{Pr}(Y=0, A=0)} = \frac{q_{10}}{q_{00}} = \frac{q_{11}}{q_{01}} = \frac{\text{Pr}(Y=1, A=1)}{\text{Pr}(Y=0, A=1)}.$$
> > > It implies that $\dfrac{q_{00}+q_{10}}{q_{10}} = \dfrac{q_{00}}{q_{10}} + 1 = \dfrac{q_{01}}{q_{11}} + 1 = \dfrac{q_{01}+q_{11}}{q_{11}}$. Hence, we get demographic parity for $y=h^{*}(x, a)$ because
> > > $$\text{Pr}(Y=1|A=0) = \frac{q_{10}}{q_{00}+q_{10}} = \frac{q_{11}}{q_{01}+q_{11}} = \text{Pr}(Y=1|A=1).$$
> > > 2. **The question I ask for the assumption for training and test IID is because some fairness issues are rooted in subpopulation shifts, for example, spurious correlation. Such a shift can result in the fair representation learning failing.**: Thank you for the clarification. Our results for finding the nearest ideal distribution can provide a recipe to correct data bias. However, what kind of data bias it can correct (e.g., subpopulation shift) and the role of non-iid sampling towards this are beyond the scope of our paper.
> > > 3. **Even after reading through the rebuttal, I still feel that Proposition 3.3 is a very strong assumption.** Proposition 3.3 provably guarantees exact fairness of the Bayes optimal classifier for downstream cost-senstitive risk. The conditions in Proposition 3.3 are both necessary and sufficient, and they naturally strengthen known fair pre-processing (Remark 3.4). So we respectfully disagree that these conditions are excessively strong.

---

> > > > ### Author Response · Authors · 2025-08-06
> > > >
> > > > We greatly appreciate your questions and feedback towards improving the presentation and clarity. We hope that our response has satisfactorily addressed your concerns, and we sincerely request you to reconsider our score. Thank you.

---

### Official Review · Reviewer_KFDe · 2025-07-02

**Clarity:** 2
**Significance:** 3
**Originality:** 4
**Rating:** 4
**Confidence:** 3

**Summary:**

This work introduces a methodology to steer a data distribution towards an ideal one, formalised as a data distribution on which a Bayes-optimal classifier satisfies exact group fairness (e.g., no DP or EO difference). The framework optimises for the closest ideal distribution via minimal KL divergence, pushing the data (or latent representations) towards it. The paper provides the conditions under which a data distribution is ideal, especially for common parametric families (Gaussian, log-normal). The authors further provide efficient solutions for interventions: an “affirmative action” intervention, adjusting the under-privileged group only), and changing all the subgroups.
Results on simulations show perfect fairness at low KL cost, and some real-world applications show bias reduction on multi-class classification by steering LLM representations, without sacrificing accuracy.
The main contributions are:
- The definition and formalization of ideal distributions without a fairness-utility trade-off
- The formulation of the problem of finding the nearest ideal distribution as a KL divergence minimization problem, with some efficient solutions through particular interventions
- Empirical results showing bias reduction in synthetic and real-world cases, while preserving accuracy.

**Questions:**

- This work is limited to common parametric families. What kind of applications still work under this assumption, and what others do not? A couple of examples would help. Although the checklist mentions that limitations are discussed in Section 7, this is not clearly stated (on the other hand, the future work is well discussed).
- The authors mention the framework of ideal distributions can also be applied to audit deployed model for fairness, particularly when the evaluation data may itself be biased. Beyond the steering showed, how can this be achieved in practice? Use the KL gap as a proxy for what’s needed to debias the model? It's not trivial to understand from the text.
- In the future work the authors mention normalising flows or mixture models as possibly integrated. Would this be a relaxation of the parametric condition (through intermediate transformations or mapping), or would this have more profound implications?
- It’s not clear if the framework would generalise beyond one binary sensitive attribute.

**Ethical Concerns:**

["NO or VERY MINOR ethics concerns only"]

**Final Justification:**

I appreciate the authors' effort in addressing my questions and concerns. There still seem to be some important open questions to be addressed in future work. Nonetheless, this is a relevant contribution to the topic, theoretically grounded but hinting at some practical applications. I'll keep my positive score.

**Limitations:**

The authors could improve their limitations by explicating stating them as such: the parametric assumptions, for instance. The practical guidance is not discussed as well. For example, larger KL gaps should be harder to overcome, but what would be the practical consequences, limits, and constraints?

**Paper Formatting Concerns:**

No major formatting issues, but an extensive revision for typos and clarity is strongly recommended.

**Quality:**

3

**Strengths And Weaknesses:**

Strengths
- Although the high-level idea of searching for an unbiased distribution has been explored theoretically before, with the existence of such distributions being proven, this work has a significant contribution by bridging a solid theoretical foundation on how to achieve such a distribution with provable fairness guarantees (even if constrained to some parametric families), and their practical application.
- The paper has a strong theoretical foundation, from the formalisation of the ideal distribution (for any cost-sensitive Bayes-optimal classifier). Several theorems and propositions provide a framework to understand the necessary and sufficient conditions for the “ideal” property (in Gaussian conditions). The optimization formulation is also well defined. Although recognised as a non-convex problem, the authors provide solutions by restricting to some parametric families. The proposed interventions are also well supported, and it is shown that the fairness/accuracy bounds depend only on the KL divergence.
- The paper demonstrates the framework’s potential in different and versatile scenarios, from synthetic data, real-world bias reduction (Bias-in-Bios dataset), and generative text-sentiment steering (for “joy”). The results are consistent and show the fairness improvements with negligible accuracy drops (or even improvements).

Weaknesses
- Not an actual weakness, but some limitations (expanded below) ar ethe parametric assumptions and the application to Bayes-optimal classifiers
- The paper is dense in some parts, and there are some notation and clarity issues. A thorough revision is suggested to improve clarity and correctness at this level, but here are some catches:
   - Typos: D’ vs D tilde, BR where BE should be (Fig 1), a “tildeh” for h tilde, etc.)
   - Abstract: initial sentence is a strong statement since other fairness angles (other than group fairness) can also be applied; and the final sentences seem repetitive
   - The optimization is for an ideal distribution, but some text parts give space to there being more of such distributions, although it is not clearly discussed.
   - In Definition 2.1 regarding DP the positive rate is mentioned. Shouldn’t it be the positive rate difference?
   - It’s not really clear in Section 6.2 how the simulation results are obtained exactly.
- Some details are not clear on the optimization procedure in practice, and for the representation steering as well (e.g., is the downstream classifier retrained or is it inference-time like the post-training intervention?). Also, the alpha value influence (e.g., it is said that excessive nudging “distorts the steering vector” but the potential reason is not discussed).

---

> ### Author Rebuttal · Authors · 2025-07-31
>
> Thank you for carefully reading our paper and providing valuable feedback. We address your concerns below.
>
> 1. **This work is limited to common parametric families. What kind of applications still work under this assumption, and what others do not? A couple of examples would help. Although the checklist mentions that limitations are discussed in Section 7, this is not clearly stated (on the other hand, the future work is well discussed).**: Our theoretical results are on the fairness of the Bayes optimal classifier and the corresponding optimization programs, which are tractable and mathematically easier to analyse for parametric families of distributions, e.g., multivariate Gaussians. Multivariate Gaussians may not always fit real-world data but can be justifiably used to model concepts and internal representations in machine learning as shown in prior work [1,2]. There can be scenarios and applications where a parametric assumption may not fit well with the input data distribution, and hence, analyzing the Bayes optimal fair classifier will become highly non-trivial. We will discuss these limitations more elaborately in future versions of the manuscript. Thank you for this feedback.
>
> 2. **The authors mention the framework of ideal distributions can also be applied to audit deployed model for fairness, particularly when the evaluation data may itself be biased. Beyond the steering showed, how can this be achieved in practice? Use the KL gap as a proxy for what’s needed to debias the model? It's not trivial to understand from the text.**: Prior work on fairness audit has mostly focused on auditing a given model instead of auditing training or evaluation data. There is a long line of work to demonstrate that the fairness-accuracy trade-off disappears when data bias is accounted for [4,5,6,7,8,9]. The KL gap in our formulation can be used as a metric of data bias and our Theorem 4.1 essentially gives an algorithmic recipe to minimally change a given data distribution to bridge this gap.
>
> 3. **In the future work the authors mention normalising flows or mixture models as possibly integrated. Would this be a relaxation .... or would this have more profound implications?**: Thisen et al. [5] showed that the Bayes error is invariant to invertible transforms, like the ones used with Normalizing flows. Furthermore, they show how to obtain an entire family of distributions using a single family of flows. The ideas presented in this paper can be used to construct invertible maps to a desired parametric family, apply our ideas of searching for ideal distributions within the available family of distributions, and then use the map back using the invertible transformation. This is our plan for future work.
>
> 4. **It’s not clear if the framework .... one binary sensitive attribute.**: Proposition 3.2 is presented for a multi-class, multiple attribute, and multivariate Gaussian setting. To show a stronger necessary and sufficient set of results, we make use of the binary label and sensitive attribute setup for Proposition 3.3. Furthermore, even for the Affirmative action intervention in Theorem 4.1, we can write down a multi-class and multiple sensitive attributes version. In fact, that is what we used to set up experiments for Section 6.1 (Multi-class debiasing) and the details of this is described in Section 4.1 of the supplementary material. One thing to note is that Affirmative action needs a non-trivial definition for situations beyond binary labels and binary sensitive attributes, since we don't have the notion of a favourable group and a positive outcome anymore. That is why we can either define Affirmative action class-wise, assuming the most "accurate" group to be favourable, or we can do it the other way around, where we first find the most favourable group and then find ideal distributions relative to that fixed group. This is what we did for the results in Section 6.1, where we worked with a multi-class, binary attribute setup, but the same setup can be perfectly used for multiple sensitive attributes.
>
> Below, we comment on some of the points raised in the Weaknesses section.
>
> 1. **The paper is dense in some .... at this level**: We will carefully review all notations and make the exposition easier in the next version of this manuscript. More discussion about the experiments in Section 6.2 is included in Section 4.2 of the Supplementary material. We will add more details in the main text in future revisions.
>
> 2. **Abstract: initial sentence is a strong .... seem repetitive**: The scope of this work is exact group fairness, and hence we wanted to be very precise in the abstract. We can refine the abstract to highlight this. Regarding the repetitiveness, we wanted to emphasize in the first sentence that we want to improve the representations used for downstream classification. In the second sentence, we wanted to express that we are also able to steer the internal representations of an LLM so that we can improve generation across groups in LLMs. We will make these clarifications in future versions of this manuscript.
>
> 3. **The optimization is for an ideal distribution, but some text parts give space to there being more of such distributions, although it is not clearly discussed.**: Our aim in Section 4 is to propose an optimization program to obtain an ideal distribution where the Bayes optimal classifier is also fair. The objective is to find the optimal ideal distribution that minimizes the KL divergence relative to the given distribution.
>
> 4. **In Definition 2.1 .... positive rate difference?**: Thank you for pointing that out. We will fix that typo in future versions of the manuscript.
>
> 5. **It’s not really clear in Section 6.2 how the simulation results are obtained exactly.**: Due to space constraints, we have laid out the details of the experiments in Section 6.2 in Section 4.2 of the supplementary material, along with all the prompts and a flow of the pipeline. To summarize, we observe a disparity in performance between groups when we measure the effectiveness of steering generation of joyful responses, using the method a recent work on modeling concepts using gaussian distributions [1]. We therefore applied our Affirmative intervention from Theorem 4.1 to obtain the set of ideal distributions for the concepts such that both groups in the data get steered effectively during inference. We use the obtained ideal distribution to nudge the optimal steering vector towards better generation for both groups.
>
> 6. **Some details are not clear on the optimization procedure in practice..... intervention?.**: For Section 6.1, the downstream classifier is not retrained. For the setup in Section 6.1, we are steering the distribution of representations towards an ideal distribution, and then training a classifier without any constraints. As indicated in Figure 3, this improves the fairness across multiple professions and for some professions, even matches and beats the Affine Steering method from Singh et al. [10].
>
> 7. **Also, the alpha value influence ... not discussed.** As stated in the paper, for the experiments in Section 6.2, alpha is required because after our intervention, we get the ideal subgroup distributions where the tradeoff between utility and fairness is alleviated. This notion of utility is related to, but different from, accuracy, for which we have ideal distribution guarantees in Theorem 4.1. Therefore, the newly obtained steering vectors can be drastically different than the optimal steering vectors (as demonstrated in Figure 4, where a high value of alpha closer to 1 puts more weight on the affirmative action vector and therefore cannot steer towards joyful generation). But we can use our obtained ideal distribution as a direction to nudge the optimal steering vector. As indicated by Figure 4, this successfully steers the text generation towards joyful sentiment for the underperforming group. We have laid out more details of the steering experiment in Section 4.2 of the supplementary material, and if you recommend, we can also add a pseudocode of the algorithm for more clarity in future revisions of the manuscript.
>
> We sincerely thank you for reading our paper carefully. We request that you kindly revisit our paper with our detailed explanations above and upgrade your rating.
>
> [1] Zhao, Haiyan, et al. "Beyond single concept vector: Modeling concept subspace in llms with gaussian distribution." arXiv preprint arXiv:2410.00153 (2024)
>
> [2] Eftekhari, Daniel, and Vardan Papyan. "On the Importance of Gaussianizing Representations." arXiv preprint arXiv:2505.00685 (2025).
>
> [3] Theisen, Ryan, et al. "Evaluating state-of-the-art classification models against bayes optimality." Advances in Neural Information Processing Systems 34 (2021): 9367-9377.
>
> [4] Wick, Michael, and Jean-Baptiste Tristan. "Unlocking fairness: a trade-off revisited." Advances in neural information processing systems 32 (2019).
>
> [5] Blum, Avrim, and Kevin Stangl. "Recovering from biased data: Can fairness constraints improve accuracy?." arXiv preprint arXiv:1912.01094 (2019).
>
> [6] Dutta, Sanghamitra, et al. "Is there a trade-off between fairness and accuracy? a perspective using mismatched hypothesis testing." International conference on machine learning. PMLR, 2020.
>
> [7] Maity, Subha, et al. "Does enforcing fairness mitigate biases caused by subpopulation shift?." Advances in Neural Information Processing Systems 34 (2021): 25773-25784.
>
> [8] Sharma, Mohit, and Amit Deshpande. "How far can fairness constraints help recover from biased data?." arXiv preprint arXiv:2312.10396 (2023).
>
> [9] Leininger, Charlotte, Simon Rittel, and Ludwig Bothmann. "Overcoming Fairness Trade-offs via Pre-processing: A Causal Perspective." arXiv preprint arXiv:2501.14710 (2025).
>
> [10] Singh, Shashwat, et al. "Representation surgery: Theory and practice of affine steering." arXiv preprint arXiv:2402.09631 (2024).

---

> > ### Comment · Reviewer_KFDe · 2025-08-05
> > **Clarification**
> >
> > Thank you for the thorough rebuttal. Most issues are now clearer, and although the theoretical foundation is sound, I'm still struggling with two practical points:
> >
> >  - Q2: I understand how the KL gap can flag bias in an evaluation set, yet I’m unsure how the steering is intended to work once the model is deployed. In a streaming or mini-batch setting, each input (or small batch) likely comes from a biased distribution. Is the idea to steer every inference sample toward the nearest ideal distribution, or is steering applied only offline to the test data? If the former, what are the implications of per-sample steering? If the latter, how should practitioners interpret evaluation metrics that rely on a distribution different from what the model will see in production? Additionally, do you have any hints on what a reasonable KL gap is to try mitigating through steering? I can imagine scenarios in which the gap is so large that the transformation loses "meaning".
> >
> >  - Q4: You note that defining “affirmative action” is non-trivial beyond the binary-label/binary-attribute case, and that one must fix a reference group. In realistic tasks with multiple sensitive attributes (or no obvious “favourable” class), pushing toward fairness for one group could hurt others, suggesting an inevitable trade-off (typical in multi-group fairness analysis). Could you briefly explain how to choose between the affirmative-action update and the change-all-subgroups setting in these scenarios? Are there heuristics, perhaps based on the observed KL gap or covariance structure, that indicate which intervention yields the best fairness–utility balance across all groups?
> >
> > Thank you,

---

> > > ### Author Response · Authors · 2025-08-06
> > > **Response to Questions**
> > >
> > > Thank you for your follow-up questions. Please find our responses below.
> > >
> > > 1a. **I understand how the KL gap can flag bias in an evaluation set, yet I’m unsure how the steering is intended to work once the model is deployed. In a streaming or mini-batch setting, each input (or small batch) likely comes from a biased distribution. Is the idea to steer every inference sample toward the nearest ideal distribution, or is steering applied only offline to the test data? If the former, what are the implications of per-sample steering? If the latter, how should practitioners interpret evaluation metrics that rely on a distribution different from what the model will see in production?** We use steering in the same way as Singh et al. (Representation Surgery: Theory and Practice of Affine Steering, ICML'24, arxiv: 2402.09631) and cite them. As an illustrative example, changing the mean and the covariance of a Gaussian distribution can be attained by applying an affine transformation (or steering function) per sample. This transformation can be computed from offline data but applied per-sample at test time. Practitioners must keep in mind that per-sample steering is more general than steering distributions, e.g., per-sample steering can change individual predictions in ways that do not change any distribution level metrics (e.g., EO, KL gap).
> > >
> > > 1b. **Additionally, do you have any hints on what a reasonable KL gap is to try mitigating through steering? I can imagine scenarios in which the gap is so large that the transformation loses "meaning".** This is an interesting future direction. As you rightly pointed out, what is a reasonable KL gap depends on how significant the fairness and accuracy gains are and whether the transformation is meaningful. In practice, one can apply distribution steering for actionable policy (e.g., changing education policy for a certain demographics to improve the fairness of their credit-worthiness).
> > >
> > > 2a. **You note that defining “affirmative action” is non-trivial beyond the binary-label/binary-attribute case, and that one must fix a reference group. In realistic tasks with multiple sensitive attributes (or no obvious “favourable” class), pushing toward fairness for one group could hurt others, suggesting an inevitable trade-off (typical in multi-group fairness analysis). Could you briefly explain how to choose between the affirmative-action update and the change-all-subgroups setting in these scenarios?** In typical fair classification, forcing a model to be fair for one group can possibly hurt others. In steering by affirmative action, where we change the data/feature distribution of only one group (EF Affirmative), it cannot worsen the performance of the optimal group-aware classifier on other groups whose distribution remains unchanged. In theory, EF Affirmative requires solving a simpler convex optimization, whereas EF-All Subgroups requires solving a harder non-convex optimization.
> > >
> > > 2b. **Are there heuristics, perhaps based on the observed KL gap or covariance structure, that indicate which intervention yields the best fairness–utility balance across all groups?** We consider this as future work. A practical heuristic based on the Rawlsian approach is as follows: Apply EF-Affirmative to the worst-off group to match its TPR-difference with that of the second worst-off group. Then improve both of them to match the third worst-off group, and so on. A gradual change might lead to better trade-offs than EF-All Subgroups. There is no silver bullet in our opinion, and we agree with your suggestion as an interesting future direction.

---

### Official Review · Reviewer_1dpz · 2025-07-03

**Clarity:** 2
**Significance:** 3
**Originality:** 3
**Rating:** 5
**Confidence:** 3

**Summary:**

The paper proposes the concept of 'ideal distribution', defined as a distribution on which any trained model that minimizes loss will have exact fairness. The paper then provides theoretical analyses of tools to find the 'nearest' ideal distribution given a real-world distribution, as well as several practical approximations. Overall, the setup aims to show how a given input distribution of data can be 'steered' towards an ideal distribution such that the model trained on this new distribution is fair. Finally, it shows the benefits of their technique by steering LLM representations, achieving good fairness scores.

**Questions:**

One big 'discussion' missing in the paper is the gap between being able to steer underlying distributions and concerns of fairness in a real-world task. This includes both explicitly acknowledging that the distribution needs to change for training and inference, and discussing the impact of access to finite data instead of distributions (expanded further in the 'Limitations' section).

Correct me if I've misunderstood the paper, but given that the underlying distribution itself is steered, this technique is much closer to fair representation learning than any 'fairness pre-processing'. Consider the perspective of how a practitioner would take advantage of this technique. When using fair pre-processing techniques, like reweighing, massaging, sampling, etc., these techniques are only applied during training to learn a fair classifier, which can then be applied directly to the original distribution during inference. But similar to fair representation learning, the method proposed in this paper also changes the distribution itself, and its 'steering' is something that'll need to be applied not just to the training set but also to the input during inference.

This is a vital clarification, in my opinion. One, I suggest the authors improve the intutions provided in the introduction by not comparing their technique to pre-processing and instead of representation learning (or please correct me in the rebuttal if I'm wrong). And two, I suggest the authors provide a better discussion of how a practitioner can take advantage of this technique in classical settings (different from representation steering in LLMs). No additional experiments needed, but a discussion of how this technique fits within the broader landscape of fairness is important.

**Ethical Concerns:**

["NO or VERY MINOR ethics concerns only"]

**Final Justification:**

I continue to maintain my positive score.

**Limitations:**

One limitation of the work is the missing link between distribution-level analysis and real-world applications, where one has access to only a finite amount of data.

Although misaligned distributions between the minority and the majority play an important role in the bias encoded in the ML model, this is not the only cause of bias. When there is a lack of enough data from the minority, even approximating the underlying distribution is error-prone, which can further exacerbate fairness concerns (and their variance).

The paper mostly focuses on distribution-level analysis, assuming access to the underlying distribution and directly steering this distribution. While they do provide empirical results with real-world datasets, a deeper analysis of this missing link and a better discussion of what's lost when you only have access to finite data would further strengthen the paper.

**Quality:**

3

**Strengths And Weaknesses:**

Quality: The paper appears technically sound to me. Most claims are well supported, and the methods used are appropriate.

Clarity: The paper is clearly written and easy to follow. My main concerns in clarity come from a lack of appropriate framing of the work in the context of existing fairness literature and a missing discussion of the same. Please see 'Questions' and 'Limitations' for more details.

Significance: I believe the results are impactful for the community, and I can imagine useful future work building on the results presented here. It takes a unique approach to fairness, and further exploration of the concept of 'ideal distributions' can be interesting for the field.

Originality: To my knowledge, the perspective of 'ideal distributions' and steering the underlying distribution to achieve fairness is novel. However, clearly there are connections to existing literature, something that I believe can benefit from better framing (see Questions and Limitations section for details).

---

> ### Author Rebuttal · Authors · 2025-07-31
>
> Thank you for carefully reading our paper and providing valuable feedback. We address your questions regarding our work in the context of existing fair classification works below. We will certainly include this discussion in future versions of this manuscript.
>
> 1. **Relation to fair representation learning and benefit to practitioners**: We study the foundational problem of finding the nearest ideal distribution to a given distribution. This has many potential applications, such as correcting training data bias, learning fair representations, steering intermediate representations for fair generation, etc. Fair pre-processing or reweighing is closer to our mathematical setup, and steering intermediate representations gives a compelling application for LLMs. Our KL optimization program can be used to measure training data bias and guide data collection policies in practice, especially when the models are trained for fairness, but the inherent bias in data is unresolved. Prior to our work, the popular strategies to correct data bias were reweighing, fair pre-processing, and collecting more data from underrepresented groups.
>
> 2. **Going beyond Distribution-level analysis and working with finite samples**: This is certainly an important future direction for us. A distributional-level analysis allows us to express the Bayes error and true positive rates (and analogously false positive rates) as functions of the distribution parameters. These are generally intractable without any distributional assumptions. In a finite-sample regime, we have high probability estimates of the moments and other distribution-dependent quantities, so our approach leads to approximate fairness guarantees. Since the scope of our paper is exact fairness, we did not include this further analysis. To go beyond distributional assumptions, we can leverage previous work that maps given data distribution to tractable, parametric families using invertible transforms that preserve Bayes error, so our results become applicable [1]. We can also assume distributions with bounded moments instead of parametric families, and still bound the Bayes error and other relevant quantities [2]. This requires a careful analysis, and is certainly a very important future direction for us.
>
> We greatly appreciate your positive feedback and request that you champion our paper for acceptance.
>
> [1] Theisen, Ryan, et al. "Evaluating state-of-the-art classification models against bayes optimality." Advances in Neural Information Processing Systems 34 (2021): 9367-9377.
>
> [2] Murti, Chaitanya, and Chiranjib Bhattacharyya. "DisCEdit: Model Editing by Identifying Discriminative Components." Advances in Neural Information Processing Systems 37 (2024): 47261-47296.

---

> > ### Comment · Reviewer_1dpz · 2025-08-03
> >
> > I appreciate the clarification. I will retain my positive score.

---

> > > ### Author Response · Authors · 2025-08-03
> > >
> > > Thank you once again for your prompt response and positive feedback.

---

### Note · Authors · 2025-08-14

We summarize the significance of our work and the important rebuttal points below with the hope that they will help you arrive at the final decision sooner.

**Significance**: In case some reviewers missed it, our work proposes a foundational framework in fair classification for steering a given (possibly biased) data distribution towards its closest ideal distribution on which the fairness-accuracy trade-off disappears. Our contributions are both theoretical and empirical, particularly significant for real-world applications such as fair pre-processing and representation steering, but not limited to these alone.
- On the theoretical side, when the given data distribution belongs to a standard parametric family, we give efficient optimization algorithms to find its nearest ideal distribution. We also study variants of this problem as the minimal affirmative action on data to guarantee both fair and accurate classification downstream.
- On the empirical side, we show the efficacy of our algorithms for steering intermediate representations towards fairness in multi-class classification and for steering LLM generation (e.g., joyful generation).

**Rebuttal Points**:
- We thank Reviewers 1dpz and KFDe for helping us improve the presentation and their suggestion of discussing multiple potential applications of our work to position it better.
- As suggested by Reviewer YLZ4, we will move an additional result from the supplementary material to the main paper.
- We thank Reviewer c89k for proofreading and the discussion on steering distributions towards ideal (i.e., no fairness-accuracy trade-off) against the inevitable trade-offs in literature when the distribution remains unchanged.
- All the reviewers made excellent suggestions regarding future directions that would help us update parts of Section 7 (Discussion and Future Work).

We greatly appreciate your help in coordinating the authors-reviewers discussion and for carefully reading our final comments.

---

### Decision · Program_Chairs · 2025-09-17

**Decision:**

Accept (poster)

**Comment:**

The authors study how to steer biased data distributions toward their closest "ideal" distribution, where "ideal" means that the distribution does not have a fairness-accuracy tradeoff. They present theoretical results for parametric families and supplement with empirical results for multi-class classification and emotion steering in LLM outputs.

Strengths:
- The paper is clearly written and well-motivated.
- The authors introduce "ideal distributions" -- something that could be useful much more broadly in fair ML.
- The authors effectively rebut many questions/limitations raised by reviewers during the review process.

Weaknesses:
- The results only apply to certain parametric families.
- The concept of a "closest ideal distribution" is insufficiently motivated. In particular, what if the "closest ideal distribution" actually has quite different downstream performance than the original one? Reviewer 1dpz brings up this point; changing the input data will lead to other downstream changes.